# Neural Guided Diffusion Bridges

**Gefan Yang** [1]  **Frank van der Meulen** [2]  **Stefan Sommer** [1]

## Abstract

We propose a novel method for simulating conditioned diffusion processes (diffusion bridges) in Euclidean spaces. By training a neural network to approximate bridge dynamics, our approach eliminates the need for computationally intensive Markov Chain Monte Carlo (MCMC) methods or score modeling. Compared to existing methods, it offers greater robustness across various diffusion specifications and conditioning scenarios. This applies in particular to rare events and multimodal distributions, which pose challenges for score-learning- and MCMC-based approaches. We introduce a flexible variational family, partially specified by a neural network, for approximating the diffusion bridge path measure. Once trained, it enables efficient sampling of independent bridges at a cost comparable to sampling the unconditioned (forward) process.

## 1. Introduction

Diffusion processes play a fundamental role in various fields such as mathematics, physics, evolutionary biology, and, recently, generative models. In particular, diffusion processes conditioned to hit a specific point at a fixed future time, which are often referred to as *diffusion bridges*, are of great interest in situations where observations constrain the dynamics of a stochastic process. For example, in generative modeling, stochastic imputation between two given images, also known as the image translation task, uses diffusion bridges to model dynamics (Zhou et al., 2024; Zheng et al., 2025). In the area of stochastic shape analysis and computational anatomy, random evolutions of biological shapes of organisms are modeled as non-linear diffusion bridges, and simulating such bridges is critical to solving inference and

registration problems (Arnaudon et al., 2022; Baker et al., 2024b; Yang et al., 2025). Additionally, diffusion bridges play a crucial role in Bayesian inference and parameter estimations based on discrete-time observations. (Delyon & Hu, 2006; van der Meulen & Schauer, 2017; 2018; Pieschner & Fuchs, 2020)

Simulation of diffusion bridges in either Euclidean space or manifolds is nontrivial since, in general, there is no closed-form expression for transition densities, which is key to constructing the conditioned dynamics via Doob's $h$-transform (Rogers & Williams, 2000). This task has gained a great deal of attention in the past decades (Beskos et al., 2006; Delyon & Hu, 2006; Schauer et al., 2017; Whitaker et al., 2016; Bierkens et al., 2021; Mider et al., 2021; Heng et al., 2022; Chau et al., 2024; Baker et al., 2024a). Among them, one common approach is to use a proposed bridge process (called *guided proposal*) as an approximation to the true bridge. Then either MCMC or Sequential Monte Carlo (SMC) methods are deployed to sample the true bridge via the tractable likelihood ratio between the true and proposed bridges. Another solution is to use the *score-matching* technique (Hyvarinen, 2005; Vincent, 2011) to directly approximate the intractable score of the transition probability using gradient-based optimization. Here, a neural network is trained with samples from the unconditioned process (Heng et al., 2022) or adjoint process (Baker et al., 2024a), and plugged into numerical solving schemes, for example, Euler-Maruyama. Several recent studies deal with the extension of bridge simulation techniques beyond Euclidean spaces to manifolds (Sommer et al., 2017; Jensen & Sommer, 2023; Grong et al., 2024; Corstanje et al., 2024). All of these rely on either a type of guided proposal or score matching.

Both guided-proposal-based and score-learning-based bridge simulation methods have certain limitations: the guided proposal requires a careful choice of a certain "auxiliary process". (Mider et al., 2021) provided various strategies, but it is fair to say that guided proposals are mostly useful when combined with MCMC or Sequential Monte Carlo (SMC) methods. In case of a strongly nonlinear diffusion or high-dimensional diffusion, the simulation of bridges using guided proposals combined with MCMC (most notably the preconditioned Crank-Nicolson (pCN) scheme (Cotter et al., 2013)) or SMC may be computationally demanding. On the other hand, score-matching relies on sampling un-

---

[1]Department of Computer Science, University of Copenhagen, Universitetsparken 1, 2100 København, Denmark [2]Department of Mathematics, Vrije Universiteit Amsterdam, De Boelelaan 1111, 1081HV Amsterdam, The Netherlands. Correspondence to: Gefan Yang <gy@di.ku.dk>.

*Proceedings of the 42ⁿᵈ International Conference on Machine Learning*, Vancouver, Canada. PMLR 267, 2025. Copyright 2025 by the author(s).

conditioned processes, and it performs poorly for bridges conditioned on rare events, as the unconditioned process rarely explores those regions, resulting in inaccurate estimation. Additionally, the canonical score-matching loss requires the inversion of $\sigma\sigma^\top$ with $\sigma$ denoting the diffusion coefficient of the process. This rules out hypo-elliptic diffusions, where $\sigma\sigma^\top$ is singular. It also poses computational challenges for high-dimensional diffusions, further exacerbating the difficulty of obtaining stable and accurate minimization of the loss.

To address these issues, we introduce a new bridge simulation method called *neural guided diffusion bridge*. It consists of the guided proposal introduced in (Schauer et al., 2017) with an additional correction drift term that is parametrized by a learnable neural network. The family of laws on path space induced by such improved proposals provides a rich variational family for approximating the law of the diffusion bridge. Once the variational approximation has been learned, independent samples can be generated at a cost similar to that of sampling the unconditioned (forward) process. The contributions of this paper are as follows:

- We propose a simple diffusion bridge simulation method inspired by the guided proposal framework, avoiding the need for score modelling or intensive MCMC or SMC updates. Once the network has been trained, obtaining independent samples from the variational approximation is trivial and computationally cheap;

- Unlike score-learning-based simulation methods, which rely on unconditional samples for learning, our method is grounded to learn directly from conditional samples. This results in greater training efficiency, especially for learning the bridges that are conditioned on rare events.

- We validate the method through numerical experiments ranging from one-dimensional linear to high-dimensional nonlinear cases, offering qualitative and quantitative analyses. Advantages and disadvantages compared to the guided proposal (Mider et al., 2021) and two score-learning-based methods, (Heng et al., 2022) and (Baker et al., 2024a), are included.

## 2. Related Work

**Diffusion bridge simulation:** This topic has received considerable attention over the past two decades and it is hard to give a short complete overview. Early contributions are (Clark, 1990; Chib et al., 2004; Delyon & Hu, 2006; Beskos et al., 2006; Lin et al., 2010; Golightly & Wilkinson, 2010). The approach of guided proposals that we use here was introduced in (Schauer et al., 2017) for fully observed uniformly

elliptic diffusions and later extend to partially observed hypo-elliptic diffusions in (Bierkens et al., 2020).

Another class of methods approximate the intractable transition density using machine learning or kernel-based techniques. (Heng et al., 2022) applied score-matching to define a variational objective for learning the additional drift in the reversed diffusion bridge. (Baker et al., 2024a) proposed learning the additional drift directly in the forward bridge via sampling from an adjoint process. (Chau et al., 2024) leveraged Gaussian kernel approximations for drift estimation.

The method we propose is a combination of existing ideas. It used the guided proposals from (Schauer et al., 2017) to construct a conditioned process, but learns an additional drift term parametrized by a neural network using variational inference.

**Diffusion Schrödinger bridge:** The diffusion bridge problem addressed in this paper may appear similar to the diffusion Schrödinger bridge (DSB) problem due to their names, but they are fundamentally different. A diffusion bridge is a process conditioned to start and end at specific points, often used for simulating individual sample paths under endpoint constraints. In contrast, a Schrödinger bridge connects two marginal distributions over time by finding the most likely stochastic process (relative to a reference diffusion) that matches these marginals. While both modify the original dynamics, the diffusion bridge imposes hard constraints on endpoints, whereas the Schrödinger bridge enforces them in distribution. Although DSB has gained attention for applications in generative modelling (Thornton et al., 2022; De Bortoli et al., 2021; Shi et al., 2024; Tang et al., 2024), it is important to recognize the distinctions between these problems.

**Neural SDE:** Neural SDEs generalize neural ODEs (Chen et al., 2018) by introducing stochasticity, enabling the modeling of systems with inherently random dynamics. Research in this area can be broadly divided into two categories: (1) modeling terminal state data (Tzen & Raginsky, 2019a;b), and (2) modeling entire trajectories (Li et al., 2020; Kidger et al., 2021). Our approach falls into the latter category, leveraging trainable drift terms and end-point constraints to capture full trajectory dynamics.

## 3. Preliminaries: Recap on Guided Proposals

### 3.1. Problem Statement

Let $(\Omega, \mathcal{F}, \mathbb{P})$ be a probability space with filtration $\{\mathcal{F}_t\}_{t\in[0,T]}$, $W$ a $d_w$-dimensional $\mathbb{P}$-Wiener process, $b : [0,T] \times \mathbb{R}^d \to \mathbb{R}^d$ and $\sigma : [0,T] \times \mathbb{R}^d \to \mathbb{R}^{d \times d_w}$ the drift- and diffusion coefficients. A $d$-dimensional $\{\mathcal{F}_t\}$-adapted diffusion process $X$ with the law of $\mathbb{P}$ is defined as the

strong solution to the stochastic differential equation (SDE):

$$\mathrm{d}X_t = b(t, X_t)\mathrm{d}t + \sigma(t, X_t)\mathrm{d}W_t, \ X_0 = x_0 \in \mathbb{R}^d. \quad (1)$$

The coefficients $b, \sigma$ are assumed to be Lipschitz continuous and of linear growth to guarantee the existence of a strong solution $X_t$ (Øksendal, 2014, Chapter 5.2). In addition, we impose the standing assumption that $X$ admits smooth transition densities $p$ with respect to the Lebesgue measure $\lambda$ on $(\mathbb{R}^d, \mathcal{B}(\mathbb{R}^d))$, where $\mathcal{B}(\mathbb{R}^d)$ is the Borel algebra of $\mathbb{R}^d$. That is, $\mathbb{P}(X_t \in A \mid X_s = x) = \int_A p(t, y \mid s, x)\lambda(\mathrm{d}y)$ for $0 \le s < t \le T, A \subset \mathbb{R}^d$.

**Notation 3.1.** Let $\mathbb{P}^\circ, \mathbb{P}^\star$ and $\mathbb{P}^\bullet$ be measures on $(\Omega, \mathcal{F})$, we denote the laws of $X$ on $\mathcal{C}([0, T], \mathbb{R}^d)$ under $\mathbb{P}^\circ, \mathbb{P}^\star$ and $\mathbb{P}^\bullet$ by $\mathbb{L}^\circ, \mathbb{L}^\star$ and $\mathbb{L}^\bullet$ respectively. For notational ease, the expectations under $\mathbb{P}^\circ, \mathbb{P}^\star$ and $\mathbb{P}^\bullet$ (and similarly $\mathbb{L}^\circ, \mathbb{L}^\star$ and $\mathbb{L}^\bullet$) are denoted by $\mathbb{E}^\circ, \mathbb{E}^\star$ and $\mathbb{E}^\bullet$ respectively. The process $X$ under $\mathbb{L}^\circ, \mathbb{L}^\star$ and $\mathbb{L}^\bullet$ is sometimes denoted by $X^\circ, X^\star$ and $X^\bullet$ respectively. For any measure $\mathbb{Q}$ on $(\Omega, \mathcal{F})$, we always denote its restriction to $\mathcal{F}_t$ by $\mathbb{Q}_t$.

The following proposition combines Proposition 4.4 and Example 4.6 in (Pieper-Sethmacher et al., 2025). It shows how the dynamics of $X$ change under observing certain events at time $T$.

**Proposition 3.2.** *Fix $t < T$. Let $y \in \mathbb{R}^d$ and $q(\cdot \mid y)$ be a probability density function with respect to a finite measure $\nu$. Let $h(t, x) = \int_{\mathbb{R}^d} p(T, y \mid t, x)q(v \mid y)\nu(\mathrm{d}y)$, and define the measure $\mathbb{P}_t^\star$ on $\mathcal{F}_t$ by $\mathrm{d}\mathbb{P}_t^\star := \frac{h(t, X_t)}{h(0, x_0)}\mathrm{d}\mathbb{P}_t$. Then under the new measure $\mathbb{P}_t^\star$, the process $X$ solves the SDE*

$$\begin{aligned} \mathrm{d}X_t &= b^\star(t, X_t)\,\mathrm{d}t + \sigma(s, X_s)\,\mathrm{d}W_s^\star, \\ b^\star(s, x) &= b(s, x) + a(s, x)r(s, x), \quad X_0 = x_0. \end{aligned} \quad (2)$$

*where $r(s, x) = \nabla_x \log h(s, x)$, $a(s, x) = \sigma(s, x)\sigma^\top(s, x)$ and $W^\star$ is a $\mathbb{P}^\star$-Wiener process.*

*Furthermore, for any bounded and measurable function $g$ and $0 \le t_1 \le ... \le t_n < T$,*

$$\begin{aligned} &\mathbb{E}^\star[g(X_{t_1}, ..., X_{t_n})] \\ &= \int_{\mathbb{R}^d} \mathbb{E}[g(X_{t_1}, ..., X_{t_n}) \mid X_T = y]\,\xi(\mathrm{d}y), \end{aligned} \quad (3)$$

*where $\xi$ is the measure defined on $(\mathbb{R}^d, \mathcal{B}(\mathbb{R}^d))$ via*

$$\xi(\mathrm{d}y) = \frac{p(T, y \mid 0, x_0)q(v \mid y)\nu(\mathrm{d}y)}{\int_{\mathbb{R}^d} p(T, y \mid 0, x_0)q(v \mid y)\nu(\mathrm{d}y)}. \quad (4)$$

*Remark* 3.3. A Bayesian interpretation of Equation (4) can be obtained by considering the following hierarchical model:

$$v \mid y \sim q(v \mid y), \quad (5a)$$
$$y \sim p(T, y \mid 0, x_0). \quad (5b)$$

Here, $y$ is considered as the parameter that gets assigned the prior density $p(T, y \mid 0, x_0)$ and $v$ is the observation. Therefore, $\xi$ is the posterior distribution of $y$ and (3) shows that sampling of the conditioned process can be done by first sampling the endpoint $x_T$ from distribution $\xi$, followed by sampling a bridge connecting $x_0$ and $x_T$.

Throughout the paper, we will consider $q(v \mid y) = \psi(v; Ly, \Sigma)$, where $L \in \mathbb{R}^{d' \times d}$ with $d' \le d$ and $L$ of full (row) rank. Here, $\psi(x; \mu, \Sigma)$ denotes the density of the $\mathcal{N}(\mu, \Sigma)$-distribution, evaluated at $x$. For example, for a two-dimensional diffusion $y = \begin{bmatrix} y_1 & y_2 \end{bmatrix}^\top$ observing only the first component $y_1$ corresponds to $L = \begin{bmatrix} 1 & 0 \end{bmatrix}$ as $Ly = y_1$. In our simulation experiments, we will assume $\Sigma = \epsilon^2 \mathbf{I}$, for a small value of $\epsilon$, which is close to observing without error. Taking $\epsilon$ strictly positive stabilizes numerical computations.

### 3.2. Guided Proposal

If $p$ were known in closed form, then the conditioned process could be directly sampled from Equation (2). This is rarely the case. For this reason, let $\tilde{X}$ be an auxiliary diffusion process that admits transition densities $\tilde{p}$ in closed form. Let $\tilde{h}(t, x) = \int_{\mathbb{R}^d} \tilde{p}(T, y \mid t, x)q(v \mid y)\nu(\mathrm{d}y)$. Define

$$E_t := \frac{\tilde{h}(t, X_t)}{\tilde{h}(0, x_0)}\exp\left(\int_0^t \frac{(\partial_s + \mathcal{A})\tilde{h}}{\tilde{h}}(s, X_s)\,\mathrm{d}s\right), \quad (6)$$

where $\mathcal{A}$ is the infinitesimal generator of the process $X$, i.e. for any $f$ in its domain $\mathcal{A}f(x) = \sum_i b_i(t, x)\partial_i f(t, x) + \frac{1}{2}\sum_{i,j} a_{ij}(t, x)\partial_{ij} f(t, x)$. Let $t < T$. Under weak conditions (see e.g. (Palmowski & Rolski, 2002, Lemma 3.1)), $\mathbb{E}[E_t] = 1$. Using $\tilde{h}$ we can define the guided proposal.

**Definition 3.4.** (Schauer et al., 2017) If we define the change of measure $\mathrm{d}\mathbb{P}_t^\circ = E_t\mathrm{d}\mathbb{P}_t$, then, under $\mathbb{P}_t^\circ$, the process $X$ solves the SDE

$$\begin{aligned} \mathrm{d}X_s &= b^\circ(s, X_s)\,\mathrm{d}s + \sigma(s, X_s)\,\mathrm{d}W_s^\circ, \\ b^\circ(s, x) &= b(s, x) + a(s, x)\tilde{r}(s, x), \quad X_0 = x_0, \end{aligned} \quad (7)$$

where $s \in [0, t]$, $\tilde{r}(t, x) = \nabla_x \log \tilde{h}(t, x)$ and $W^\circ$ is a $\mathbb{P}_t^\circ$-Wiener process. The process $X$ under the law $\mathbb{P}_t^\circ$ is known as the guided proposal.

Intuitively, $X^\circ$ is constructed to resemble the true conditioned process $X^\star$ by replacing $r$ by $\tilde{r}$. Crucially, as its drift and diffusion coefficients are known in closed form, the guided proposal can be sampled using efficient numerical SDE solvers such as Euler-Maruyama.

The definition of the guided process can be extended to $[0, T]$ by continuity. Whereas $\mathbb{P}_t \ll \mathbb{P}_t^\circ$ for $t < T$ the measures will typically be singular in the limit $t \uparrow T$. Nevertheless, $\mathbb{P}_t^\star \ll \mathbb{P}_t^\circ$ may still hold under this limiting operation

and this is what matters for our purposes. In (Schauer et al., 2017) and (Bierkens et al., 2020), precise conditions are given under which $\mathbb{P}_T^\star \ll \mathbb{P}_T^\circ$. In the case of conditioning on the event $\{LX_T = v\}$ –so there is no noise on the observation– this is subtle. We postpone a short discussion on this to Section 3.4 to argue that all numerical examples considered in Section 5 will not break down in case the noise level on the observation, $\epsilon$, tends to zero. We then get the following theorem from (Bierkens et al., 2020, Theorem 2.6) that states the change of laws from $\mathbb{L}^\circ$ to $\mathbb{L}^\star$.

**Theorem 3.5.** *If certain assumptions (Bierkens et al., 2020, Assumptions 2.4, 2.5) hold, then*

$$\frac{\mathrm{d}\mathbb{L}^\star}{\mathrm{d}\mathbb{L}^\circ}(X) = \frac{\tilde{h}(0, x_0)}{h(0, x_0)}\Psi_T(X), \tag{8}$$

*where*

$$\Psi_T(X) = \exp\left(\int_0^T \frac{(\partial_t + \mathcal{A})\tilde{h}}{\tilde{h}}(s, X_s)\mathrm{d}s\right). \tag{9}$$

### 3.3. Guided Proposal Induced by Linear Process

The choice of the auxiliary process $\tilde{X}$, which determines $\tilde{h}$ and hence the guided process, offers some flexibility, as long as the conclusion of Theorem 3.5 applies. We now specialize to the case where the process $\tilde{X}$ solves a linear SDE, as in this case $\tilde{h}$ can be obtained by solving a finite-dimensional system of ordinary differential equations (ODEs). Specifically, we assume $\tilde{X}$ solves

$$\begin{aligned} \mathrm{d}\tilde{X}_t &= \tilde{b}(t, \tilde{X}_t)\,\mathrm{d}t + \tilde{\sigma}(t)\,\mathrm{d}W_t, \\ \tilde{b}(t, x) &= \beta(t) + B(t)x, \quad \tilde{X}_0 = x_0. \end{aligned} \tag{10}$$

Let $\tilde{A}$ denote the infinitesimal generator of the process $\tilde{X}$. Since $\tilde{h}$ solves $(\partial_t + \tilde{A})\tilde{h} = 0$, we can replace $(\partial_t + \mathcal{A})\tilde{h}$ by $(\mathcal{A} - \tilde{A})\tilde{h}$. This gives

$$\Psi_t(X) = \exp\left(\int_0^t G(s, X_s)\mathrm{d}s\right), \quad t \leq T, \tag{11}$$

$$G(s, x) := \left\langle b(s, x) - \tilde{b}(s, x), \tilde{r}(s, x)\right\rangle$$

$$-\frac{1}{2}\mathrm{tr}\left([a(s, x) - \tilde{a}(s)]\left[\tilde{H}(s) - (\tilde{r}\tilde{r}^\top)(s, x)\right]\right). \tag{12}$$

Here, $\tilde{H}(s)$ is the negative Hessian of $\log\tilde{h}(s, x)$, which turns out to be independent of $x$, $\tilde{a}(s) = (\tilde{\sigma}\tilde{\sigma}^\top)(s)$. Under the choice of $q(v \mid y) = \psi(v; Ly, \Sigma)$ and $\tilde{X}$, $\tilde{H}$ and $\tilde{r}$ are given by

$$\tilde{H}(t) = L^\top(t)M(t)L(t), \tag{13}$$

$$\tilde{r}(t, x) = L^\top(t)M(t)(v - u(t) - L(t)x), \tag{14}$$

where $M(t) = (M^\dagger(t))^{-1}$ and $L$, $M^\dagger$ and $u$ satisfy the system of backward ODEs (See (Mider et al., 2021, Theorem

2.4)):

$$\mathrm{d}L(t) = -L(t)B(t)\,\mathrm{d}t, \quad L(T) = L, \tag{15a}$$

$$\mathrm{d}M^\dagger(t) = -L(t)\tilde{a}(t)L^\top(t)\,\mathrm{d}t, \quad M^\dagger(T) = \Sigma, \tag{15b}$$

$$\mathrm{d}u(t) = -L(t)\beta(t)\,\mathrm{d}t, \quad u(T) = 0. \tag{15c}$$

*Remark* 3.6. A simple choice of $\tilde{X}$ is a scaled Brownian motion, i.e. $\tilde{X}_t = \tilde{\sigma}W_t$. If $\tilde{\sigma}\tilde{\sigma}^\top$ is invertible, then

$$\nabla_x \log\tilde{p}(T, v \mid s, x) = (\tilde{\sigma}\tilde{\sigma}^\top)^{-1}\frac{v - x}{T - s}. \tag{16}$$

Therefore, the guided proposal has drift $b(s, x) + \sigma(s, x)\sigma^\top(s, x)(\tilde{\sigma}\tilde{\sigma}^\top)^{-1}(v - x)/(T - s)$. If $\sigma(s, x) = \tilde{\sigma}$ this reduces to the guiding term proposed in (Delyon & Hu, 2006).

### 3.4. Choice of Linear Process

The linear process is defined by the triplet of functions $(\beta, B, \tilde{\sigma})$. In choosing this triplet, two considerations are of importance:

1. In case of conditioning on the event $\{LX_T = v\}$ –so no extrinsic noise on the observation– the triplet needs to satisfy certain "matching conditions" (see (Bierkens et al., 2020, Assumption 2.4)) to ensure $\mathbb{P}^\star \ll \mathbb{P}^\circ$. For uniformly elliptic diffusions, this only affects $\tilde{\sigma}$. In case $L = \mathbf{I}_d$, so the conditioning is on the full state, $\tilde{\sigma}$ should be chosen such that $\tilde{a}(T) = a(T, x_T)$. Hence, for this setting, we can always ensure absolute continuity. For the partially observed case, it is necessary to assume that $a$ is of the form $a(t, x) = s(t, Lx)$ for some matrix values map $s$. In that case, it suffices to choose $\tilde{a}$ such that $L\tilde{a}(T)L^\top = Ls(T, v)L^\top$. In case the diffusivity does not depend on the state, a natural choice is to take $\tilde{\sigma} = \sigma$ to guarantee absolute continuity.

   For hypo-elliptic diffusions, the restrictions are a bit more delicate. On top of conditions on $\tilde{\sigma}$, it is also required to match certain properties in the drift by choice of $B$. With the exception of the FitzHugh-Nagumo (FHN) model studied in Section 5.3, in all examples that we consider in Section 5 we have ensured these properties are satisfied. The numerical simulation results for FHN model presented in (Bierkens et al., 2020) strongly suggest that the conditions posed in that paper are actually more stringent than needed for absolute continuity. For this reason, we chose the auxiliary process just as in (Bierkens et al., 2020).

2. Clearly, the closer $\tilde{b}$ to $b$ and $\tilde{a}$ to $a$, the more the guided proposal resembles the true conditioned process. This can for instance be seen from $\log\Psi_T(X) = $

$\int_0^T \frac{(\mathcal{A}-\tilde{\mathcal{A}})\tilde{h}}{\tilde{h}}(s, X_s)\mathrm{d}s$, which vanishes if the coefficients are equal. As proposed in (Mider et al., 2021), a practical approach is to compute the first-order Taylor expansion at the point one conditions on, i.e., $\beta(t) = b(t, v)$, $B(t)x = J_b(t, v)(x - v)$, where $J_b$ is the Jacobian of $b(t, x)$ with respect to $x$. Compared to simply taking a scaled Brownian motion, this choice can result in a guided proposal that better mimics true conditioned paths.

### 3.5. Strategies for Improving upon Guided Proposals

Although the guided proposal takes the conditioning into account, its sample paths may severely deviate from true conditioned paths. This may specifically be the case for strong nonlinearity in the drift or diffusivity. There are different ways of dealing with this.

- If we write the guided path as functional of the driving Wiener process, one can update the driving Wiener increments using the pCN within a Metropolis-Hastings algorithm. Details are provided in Appendix A.1, see also the discussion in (Mider et al., 2021).

- Devising better choices for $B$, $\beta$ and $\tilde{\sigma}$.

- Adding an extra term to the drift of the guided proposal by a change of measure, where a neural network parametrizes this term. We take this approach here and further elaborate on it in the upcoming section.

## 4. Methodology

### 4.1. Neural Guided Diffusion Bridge

For a specific diffusion process, it may be hard to specify the maps $B$, $\beta$ and $\tilde{\sigma}$, which may lead to a guided proposal whose realizations look rather different from the actual conditioned paths. For this reason, we propose to adjust the dynamics of the guided proposal by adding a learnable bounded term $\sigma(s, x)\vartheta_\theta(s, x)$ to the drift. Specifically, let $\vartheta_\theta \colon [0, T] \times \mathbb{R}^d \to \mathbb{R}^d$ be a function parameterized by $\theta \in \Theta$, where $\Theta$ denotes the parameter space. Define:

$$\varkappa_t := \exp\left(\int_0^t \vartheta_\theta^\top(s, X_s)\mathrm{d}W_s^\circ - \frac{1}{2}\int_0^t \|\vartheta_\theta(s, X_s)\|^2\mathrm{d}s\right). \tag{17}$$

We impose the following assumption on $\vartheta_\theta$:

**Assumption 4.1.** The map $\vartheta_\theta$ is bounded and $x \mapsto \vartheta_\theta(s, x)$ is Lipschitz continuous, uniformly in $s \in [0, T]$.

The Lipschitz continuity ensures $\varkappa_t$ is a martingale with $\mathbb{E}[\varkappa_T] = 1$. Define a new probability measure $\mathbb{P}^\bullet$ on $(\Omega, \mathcal{F}_T)$ by

$$\mathrm{d}\mathbb{P}^\bullet := \varkappa_T \mathrm{d}\mathbb{P}^\circ \tag{18}$$

Then by Girsanov's theorem, the process $W_t^\bullet := W_t^\circ - \int_0^t \vartheta_\theta(s, X_s)\mathrm{d}s$ is a $\mathbb{P}^\bullet$-Wiener process. We now define a new diffusion process $X^\bullet$ under $\mathbb{P}^\bullet$:

**Definition 4.2.** The neural guided diffusion bridge is a diffusion process that is defined as the strong solution to the SDE:

$$\mathrm{d}X_t = b_\theta^\bullet(t, X_t)\,\mathrm{d}t + \sigma(t, X_t)\,\mathrm{d}W_t^\bullet,$$
$$b_\theta^\bullet(s, x) = b^\circ(s, x) + \sigma(s, x)\vartheta_\theta(s, x), \quad X_0 = x_0, \tag{19}$$

where $b^\circ(s, x) = b(s, x) + a(s, x)\tilde{r}(s, x)$ is defined in Equation (7).

A unique strong solution $X^\bullet$ to Equation (19) is guaranteed due to Assumption 4.1.

We propose to construct $\vartheta_\theta$ as a learnable neural network, whose goal is to approximate the difference of drift coefficients. When $\vartheta_\theta = \sigma^\top(r - \tilde{r})$, the discrepancy between $\mathbb{P}^\bullet$ and $\mathbb{P}^\star$ vanishes. Lipschitz continuity of the neural net can be achieved by employing sufficiently smooth activation functions and weight normalization. Gradient clipping can prevent extreme growth on $x$. In our numerical experiments, we use either tanh or LipSwish (Chen et al., 2019) activations and gradient clipping by the norm of $1.0$ to fulfill such conditions.

To learn the map $\vartheta_\theta$, we propose a loss function derived from a variational approximation where the set of measures $\{\mathbb{P}_\theta^\bullet; \theta \in \Theta\}$ provides a variational class for approximating $\mathbb{P}^\star$. The following theorem shows that minimizing $\theta \mapsto \mathrm{D}_{\mathrm{KL}}(\mathbb{P}_\theta^\bullet \| \mathbb{P}^\star)$ is equivalent to minimizing $L$ as defined below.

**Theorem 4.3.** *If we define the loss function by*

$$L(\theta) := \mathbb{E}^\bullet \int_0^T \left\{ \frac{1}{2}\|\vartheta_\theta(s, X_s)\|^2 - G(s, X_s) \right\}\mathrm{d}s, \tag{20}$$

*then*

$$\mathrm{D}_{\mathrm{KL}}(\mathbb{P}_\theta^\bullet \| \mathbb{P}^\star) = L(\theta) - \log\frac{\tilde{h}(0, x_0)}{h(0, x_0)}, \tag{21}$$

*with $G(s, x)$ as defined in Equation (12).*

The proof is given in Appendix A.2. Note that under $\mathbb{P}^\bullet$, the law of $X$ depends on $\theta$ and therefore the dependence of $L$ on $\theta$ is via both $\vartheta_\theta$ and the samples from $X$ under $\theta$-parameterized $\mathbb{P}^\bullet$. If $\theta_{\mathrm{opt}}$ is a local minimizer of $L$ and $L(\theta_{\mathrm{opt}}) = \log\frac{\tilde{h}(0, x_0)}{h(0, x_0)}$, then $\theta_{\mathrm{opt}}$ is a global minimizer. This implies $\mathrm{D}_{\mathrm{KL}}(\mathbb{P}_{\theta_{\mathrm{opt}}}^\bullet \| \mathbb{P}^\star) = 0$ from which we obtain $\mathbb{P}_{\theta_{\mathrm{opt}}}^\bullet = \mathbb{P}^\star$. (Heng et al., 2022) applied a similar variational formulation. However, contrary to our approach, in this work the drift of the bridge proposal is learned from unconditional forward samples. Not surprisingly, this can be inefficient when conditioning on a rare event.

It can be seen that $L$ is low bounded by $\log \frac{\tilde{h}(0,x_0)}{h(0,x_0)}$. In general, $h$ is not known in closed form, but in some simple settings it is. In such settings, we can directly inspect the value of the lower bound and assess whether the trained neural network is optimal. In Section 5.1, we provide examples for such sanity checks.

## 4.2. Reparameterization and Numerical Implementation

Optimizing $L$ by gradient descent requires sampling from a parameterized distribution $\mathbb{P}^{\bullet}$ and backpropagating the gradients through the sampling. To estimate the gradient, We use the reparameterization trick proposed in (Kingma & Welling, 2022). Specifically, the existence of a strong solution $X^{\bullet}$ to Equation (19) means that there is a measurable map $\phi_\theta : \mathcal{C}([0,T], \mathbb{R}^{d_w}) \to \mathcal{C}([0,T], \mathbb{R}^d)$, such that $X^{\bullet} = \phi_\theta(W^{\bullet})$. Here, we have dropped the dependence of $\phi_\theta$ on the initial condition $x_0$ as it is fixed throughout. The objective Equation (20) can be then rewritten as:

$$L(\theta) = \mathbb{E}^{\bullet} \int_0^T \left\{ \frac{1}{2} \|\vartheta_\theta(t, \phi_\theta(W_t))\|^2 - G(t, \phi_\theta(W_t)) \right\} \mathrm{d}t. \tag{22}$$

Choose a finite discrete time grid $\mathcal{T} := \{t_m\}_{m=0,1,\ldots,M}$, with $t_0 = 0, t_M = T$. Let $X^{\bullet}_{t_m}, W^{\bullet}_{t_m}$ be the evaluations of $X^{\bullet}, W^{\bullet}$ at $t = t_m$ respectively, $\{x^{\bullet(n)}_{t_m}\}, \{w^{\bullet(n)}_{t_m}\}, n = 1,\ldots,N$ be collections of samples of $X^{\bullet}_{t_m}, W^{\bullet}_{t_m}$, and $x^{\bullet(n)}_{t_0} = \phi_\theta(w^{\bullet(n)}_{t_0}) = x_0$. Then Equation (22) can be approximated by the Monte Carlo approximation:

$$L(\theta) \approx \frac{1}{N} \sum_{n=1}^{N} \sum_{m=1}^{M} \left\{ \frac{1}{2} \|\vartheta_\theta(t_{m-1}, \phi_\theta(w^{\bullet(n)}_{t_{m-1}}))\|^2 \right.$$
$$\left. - G(t_{m-1}, \phi_\theta(w^{\bullet(n)}_{t_{m-1}})) \right\} \delta t. \tag{23}$$

In practice, $x^{\bullet(n)}_{t_m} = \phi_\theta(w^{\bullet(n)}_{t_m})$ is implemented as a numerical SDE solver $f_\theta(w^{\bullet(n)}_{t_m}, t_{m-1}, x^{\bullet(n)}_{t_{m-1}})$ that takes the previous step $(t_{m-1}, x^{\bullet(n)}_{t_{m-1}})$ as additional arguments. As $x^{\bullet(n)}_{t_{m-1}}$ also depends on $\theta$, the gradient with respect to $\theta$ needs to be computed recursively. Leveraging automatic differentiation frameworks, all gradients can be efficiently recorded in an acyclic computational graph during the forward integration, enabling the backpropagation for updating $\theta$. While the complexity of backpropagation scales linearly with $M$ and quadratically with $d$—a property inherent to gradient-based optimization methods—our approach remains highly efficient for moderate-dimensional problems and provides a robust foundation for further scalability improvements. Further details about the gradient computation can be found in Appendix A.3 and the numerical algorithm is shown as Algorithm 1.

---

**Algorithm 1** Neural guided bridge training

**Input:** Discrete time grid $\mathcal{T} := \{t_m\}_{m=0,1,\ldots,M}$, initial $\theta$, number of iterations $K$
Solve Equation (15) on $\mathcal{T}$ backwards, obtain and store $\{\tilde{H}(t_m)\}, \{\tilde{r}(t_m, \cdot)\}$ using Equations (13) and (14).
**repeat**
  **for** $n = 1, \ldots, N$ **do**
    Sample $w^{\bullet(n)} = \{w^{\bullet(n)}_{t_m}\}$ on $\mathcal{T}$.
    Solve Equation (19) on $\mathcal{T}$ with $w^{\bullet(n)} = \{w^{\bullet(n)}_{t_m}\}$, obtain $\{x^{\bullet(n)}_{t_m}\}$.
  **end for**
  Approximate $L(\theta)$ by Equation (23).
  Backpropagate $\nabla_\theta L(\theta)$ and update $\vartheta_\theta$ by gradient descent.
**until** Iteration count $> K$

---

## 5. Experiments

### 5.1. Linear Processes

We consider one-dimensional linear processes with analytically tractable conditional drifts, including Brownian motion with constant drift and the Ornstein–Uhlenbeck process. For these models, the lower bound of $\theta \mapsto L(\theta)$ can be explicitly computed, serving as a benchmark to assess whether the neural network reaches this bound. Additional details are provided in Appendix B.2.

**Brownian bridge:** Consider a one-dimensional Brownian motion with constant drift: $\mathrm{d}X_t = \gamma \mathrm{d}t + \sigma \mathrm{d}W_t$. As its transition density $p(t, x_t \mid s, x_s)$ is Gaussian, the fully-observed process conditioned on $\{X_T = v\}$, satisfies the SDE:

$$\mathrm{d}X^{\star}_t = \frac{v - X^{\star}_t}{T - t} \mathrm{d}t + \sigma \mathrm{d}W_t. \tag{24}$$

We construct the guided proposal using the auxiliary process $\tilde{X}_t = \sigma W_t$. It is easy to see that $X^{\bullet}$ solves the SDE:

$$\mathrm{d}X^{\bullet}_t = \left\{ \gamma + \frac{v - X^{\bullet}_t}{T - t} + \sigma \vartheta_\theta(t, X^{\bullet}_t) \right\} \mathrm{d}t + \sigma \mathrm{d}W_t. \tag{25}$$

By comparing $X^{\bullet}$ with $X^{\star}$, it is clear that the optimal map $\vartheta$ is given by $\vartheta_{\theta_{\mathrm{opt}}}(t, x) = -\gamma/\sigma$. Additionally, the lower bound on $L$, $\log \frac{\tilde{h}(0,x_0)}{h(0,x_0)}$, is analytically tractable since the transition densities $\tilde{p}$ of $\tilde{X}$ are Gaussian. In Figure 6a, we track how the training varies over iterations under different settings of $\gamma$ and $\sigma$, which leads to different lower bounds. It can be seen that all the trainings converge to corresponding theoretical lower bounds. Figure 2 compares the trained map $\vartheta_\theta$ with the optimal map $\vartheta_{\theta_{\mathrm{opt}}}$. The neural network matches the optimal map in regions well supported by the training data, but the approximation error grows outside these regions—a common limitation of neural network training. Figure 3 shows the empirical marginal densities of the neural bridge alongside the analytical densities obtained from

independent simulations of Equations (24) and (25). The learned and analytical distributions are in close agreement.

**Ornstein-Uhlenbeck bridge:** We now consider the Ornstein-Uhlenbeck (OU) process:

$$dX_t = \gamma(\mu - X_t)dt + \sigma dW_t, \qquad (26)$$

which requires a $t, x$-dependent $\vartheta_{\theta_{\mathrm{opt}}}$ to correct the guided proposal. Upon choosing $\tilde{X}_t = \sigma W_t$, the neural bridge satisfies the SDE:

$$dX_t^\bullet = \left\{ \gamma(\mu - X_t^\bullet) + \frac{v - X_t^\bullet}{T - t} + \sigma \vartheta_\theta(t, X_t^\bullet) \right\} dt + \sigma dW_t. \qquad (27)$$

As with Brownian motion, the OU process has Gaussian transition densities and is therefore analytically tractable. Using this property, we derive the optimal map $\vartheta_{\theta_{\mathrm{opt}}}$ and the corresponding lower bound on $L$, given in Equations (46) and (47). We vary the parameters $\gamma$, $\mu$, and $\sigma$ and plot the resulting training-loss curves alongside the analytical lower bound in Figure 6b; in every case the loss converges to its respective bound. Figure 4 compares the outputs of the trained network with the optimal map, and Figure 5 shows the empirical marginal densities of the learned neural bridge with those of the analytical bridge obtained from independent simulations of Equations (27) and (44).

From the above two experiments, we conclude that the neural network is able to learn the optimal drift with the proposed loss function and that the neural guided bridge is very close to the true bridge in terms of KL divergence (reflected by the very small differences between training losses and analytical lower bounds). In the remaining numerical examples no closed form expression is available for $h(0, x_0)$ and therefore performance can only be assessed qualitatively.

### 5.2. Cell Diffusion Model

(Wang et al., 2011) introduced a model for cell differentiation which serves as a test case for diffusion bridge simulation in (Heng et al., 2022; Baker et al., 2024a). Cellular expression levels $X_t = \begin{bmatrix} X_{t,1} & X_{t,2} \end{bmatrix}^\top$ are governed by the 2-dimensional SDE:

$$dX_t = \begin{bmatrix} \frac{X_{t,1}^4}{2^{-4} + X_{t,1}^4} + \frac{2^{-4}}{2^{-4} + X_{t,2}^4} - X_{t,1} \\ \frac{X_{t,2}^4}{2^{-4} + X_{t,2}^4} + \frac{2^{-4}}{2^{-4} + X_{t,1}^4} - X_{t,2} \end{bmatrix} dt + \sigma dW_t, \quad (28)$$

driven by a 2-dimensional Wiener process $W$. The highly nonlinear drift makes this a challenging case. The guided neural bridge is constructed as an OU process:

$$d\tilde{X}_t = \{\mathbf{1} - \tilde{X}_t\}dt + \sigma dW_t, \qquad (29)$$

with $\mathbf{1} = \begin{bmatrix} 1 & 1 \end{bmatrix}^\top$. We study three representative fully-observed cases that result in distinct dynamics for the conditional processes: (1) events that are likely under the forward

process, which we refer to as "normal" events; (2) rare events; and (3) events that cause trajectories to exhibit multiple modes, where the marginal probability at certain times is multimodal. We compare our method to (a) the *guided proposal* (Schauer et al., 2017); (b) *bridge simulation via score matching* (Heng et al., 2022); and (c) *bridge simulation using adjoint processes* (Baker et al., 2024a). The topleft panel of Figure 1 shows realisations of forward samples of the process. The other panels show performance of the neural guided bridge upon conditioning on various events, that we detail below. Further details on these experiments are presented in Appendix B.3.

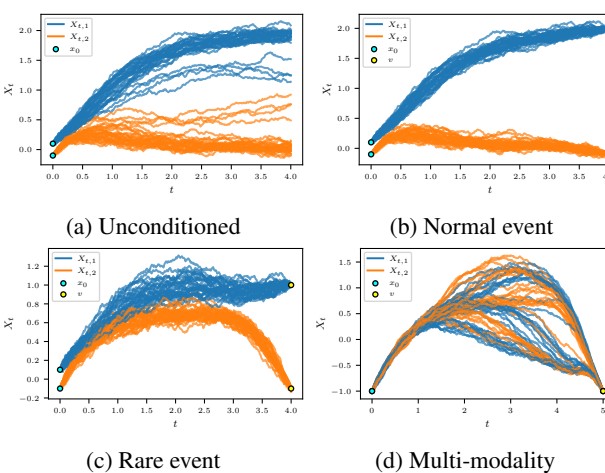

(a) Unconditioned      (b) Normal event

(c) Rare event      (d) Multi-modality

*Figure 1.* Top left (a): 30 realisations from the unconditioned cell diffusion model (Cf. Equation (28)). Top right (b): 30 realisations from the learned conditional process on a "normal event" $v = \begin{bmatrix} 2.0 & -0.1 \end{bmatrix}^\top$; Bottom left (c): 30 realisations from the learned conditional process on a rare event $v = \begin{bmatrix} 1.0 & -0.1 \end{bmatrix}^\top$; Bottom right (d): 30 realisations from the learned conditional process on event $v = \begin{bmatrix} -1.0 & -1.0 \end{bmatrix}^\top$ that causes multi-modality. Except in panel (d), all processes start from $x_0 = \begin{bmatrix} 0.1 & -0.1 \end{bmatrix}^\top$ and run up to time $T = 4.0$. In (d), the process starts from $x_0 = \begin{bmatrix} -1.0 & -1.0 \end{bmatrix}^\top$ and runs up to time $T = 5.0$.

In all the experiments in the following we take $\sigma = 0.1$.

**Normal event:** We set $x_0 = \begin{bmatrix} 0.1 & -0.1 \end{bmatrix}^\top$, $T = 4$ and $v = \begin{bmatrix} 2.0 & -0.1 \end{bmatrix}^\top$. From the topleft panel in Figure 1 it is clear that this corresponds to a normal event: balls around $v$ at time $T$ get non-negligible mass under the unconditioned process. In Figure 1b, we show 30 sample paths obtained from sampling the trained neural bridge and Figure 7 shows a comparison to the three baseline methods mentioned above. Since the true conditional process is analytically intractable, we generated $100,000$ samples from the forward (unconditional) process Equation (28), and obtained 172 samples that satisfy $\|LX_T - v\| \le 0.01$, and only show first 30 samples in the figure. Those samples can be treated as samples close to true bridges. Overall, all four methods successfully recover the true dynamics. The perfor-

mance of all four methods considered is comparable. Note that the adjoint bridge sample paths appear slightly more dispersed.

**Rare event:** We set $x_0 = \begin{bmatrix} 0.1 & -0.1 \end{bmatrix}^\top$ and $T = 4$ as before but now we take $v = \begin{bmatrix} 1.0 & -0.1 \end{bmatrix}^\top$, which is a rare event. Unlike the "normal event" case, the true dynamics cannot be recovered by forward sampling from the unconditioned process, as it is highly improbable that paths end up in a small ball around $v$. In Figure 1c, we show 30 sample paths obtained from sampling the trained neural bridge and Figure 7 shows a comparison to the three baseline methods mentioned above. All methods except the adjoint forward approach capture the correct trajectory dynamics, consistent with findings in (Baker et al., 2024a). Among the remaining three, trajectories from the neural bridge align more closely with those from the guided proposal than those from score matching. In the score matching method, samples of $X_{t,1}$ show higher variance prior to $t = 3.5$, suggesting less accurate score estimates. This is expected, as score matching learns its drift from samples of the unconditioned process which rarely reaches small balls around $v$, leading to poor approximation quality.

**Multi-modality:** In both of the previous cases, each component's marginal distribution at times $(0, T]$ is unimodal. However, with some special initial conditions, multimodality can arise, which poses a challenging task where one would like to recover all modes. Specifically, let $x_0 = \begin{bmatrix} -1.0 & -1.0 \end{bmatrix}^\top$, $T = 5.0$ and $v = \begin{bmatrix} -1.0 & -1.0 \end{bmatrix}^\top$. In Figure 1d, we show 30 sample paths obtained from sampling the trained neural bridge and Figure 7 shows a comparison to the three baseline methods mentioned above. Both the adjoint bridge and score matching fail to model the dynamics accurately. In contrast, the neural bridge and guided proposal yield similar marginal distributions. However, good performance of the guided proposal may take many MCMC iterations, or possibly the use of multiple (interacting) chains. Once trained, the neural bridge can generate independent samples, at a cost comparable to unconditioned forward simulations, while maintaining sampling quality close to that of the guided proposal.

Across all three tasks, the neural bridge shows strong adaptability and achieves performance comparable to the guided proposal, with the added benefit of faster independent sampling. In contrast, the other two methods exhibit limitations under specific settings.

### 5.3. FitzHugh-Nagumo Model

We consider the FitzHugh-Nagumo model, which is a prototype of an excitable system, considered for example in (Ditlevsen & Samson, 2019; Bierkens et al., 2020). It is described by the SDE:

$$\mathrm{d}X_t = \left\{ \begin{bmatrix} \frac{1}{\chi} & -\frac{1}{\chi} \\ \gamma & -1 \end{bmatrix} X_t + \begin{bmatrix} \frac{s - X_{t,1}^3}{\chi} \\ \alpha \end{bmatrix} \right\} \mathrm{d}t + \begin{bmatrix} 0 \\ \sigma \end{bmatrix} \mathrm{d}W_t$$

$$(30)$$

We condition the process by the value of its first component by setting $L = \begin{bmatrix} 1 & 0 \end{bmatrix}$ and hence condition on the event $LX_T = v \in \mathbb{R}$. We construct the guided proposal as proposed in (Bierkens et al., 2020) using the Taylor expansion $-x^3 \approx 2v^3 - 3v^2 x$ at $x = v$. Thus, $\tilde{X}$ satisfies the SDE

$$\mathrm{d}\tilde{X}_t = \left\{ \begin{bmatrix} \frac{1 - 3v^3}{\chi} & -\frac{1}{\chi} \\ \gamma & -1 \end{bmatrix} \tilde{X}_t + \begin{bmatrix} \frac{2v^3 + s}{\chi} \\ \alpha \end{bmatrix} \right\} \mathrm{d}t + \begin{bmatrix} 0 \\ \sigma \end{bmatrix} \mathrm{d}W_t$$

$$(31)$$

Suppose $\{\chi, s, \gamma, \alpha, \sigma\} = \{0.1, 0, 1.5, 0.8, 0.3\}$. We examine the conditional behaviour of $X$ within $t \in [0, 2.0]$ under two scenarios: (1) conditioning on a normal event $v = -1.0$; and (2) conditioning on a rare event $v = 1.1$. As the score-matching and adjoint-process methods have not been proposed in the setting of a partially observed state, we only compare our method to the guided proposal.

**Normal event:** Figure 11 compares sample trajectories generated by the neural bridge and guided proposal. As a reference, paths obtained by unconditional sampling are added, where any path not ending close to the endpoint has been rejected. Both methods capture the key features of the conditioned dynamics, closely matching the reference trajectories. However, the neural bridge achieves this using independently drawn samples after training, whereas the guided proposal requires a long MCMC chain. The similarity between the neural bridge and the guided proposal confirms that the neural bridge effectively approximates the conditioned distribution, indicating accurate modeling of the dynamics under normal event conditioning.

**Rare event:** In Figure 12 we redo the experiment under conditioning on a rare event. Both the neural bridge and the guided proposal successfully capture the bimodal structure in the trajectories, as reflected by the two distinct clusters in the $X_{t,1}$ paths, particularly evident after $t \approx 0.5$. However, the guided proposal matches the reference distribution more closely, especially in the concentration and spread of trajectories beyond time $t = 1.5$.

### 5.4. Stochastic Landmark Matching

Finally, we present a high-dimensional stochastic nonlinear conditioning task: stochastic landmark matching, see for instance (Arnaudon et al., 2022). We consider a stochastic model that describes the evolution of $n$ distinct landmarks in $\mathbb{R}^d$ of a closed nonintersecting curve in $\mathbb{R}^d$. The state at time $t$, $X_t$, consists of the concatenation of each of the $n$ landmark locations at time $t$. Hence, $X_t$ takes values in $\mathbb{R}^{dn}$. The stochastic landmark model defined in (Arnaudon et al., 2022), for ease of exposition considered without mo-

| | OU ($d = 1$) | | Cell ($d = 2$) | | FHN ($d = 2$) | | Landmark ($d = 20$) | |
|---|---|---|---|---|---|---|---|---|
| Methods | #Params | Time | #Params | Time | #Params | Time | #Params | Time |
| Adjoint forward (Baker et al., 2024a) | $21,969$ | $162.68s$ | $22,114$ | $265.06s$ | N/A | N/A | $114,744$ | $543.80s$ |
| Score matching (Heng et al., 2022) | $26,353$ | $65.55s$ | $26,498$ | $103.38s$ | N/A | N/A | $26,766$ | $353.05s$ |
| Neural guided bridge (ours) | $921$ | $44.12s$ | $2,306$ | $94.79s$ | $3,362$ | $113.77s$ | $15,188$ | $122.48s$ |

*Table 1.* Benchmarks with two other deep-learning based methods. Neither adjoint forward nor score matching is available for the FHN partially observed conditioning case (denoted as "N/A", not applicable in the table). All the experiments are conducted with the parameters described in Appendix B.

mentum variables, defines the process $(X_t, t \in [0, T])$ as the solution to the SDE

$$\mathrm{d}X_t = Q(X_t)\mathrm{d}W_t, \quad Q(X_t)_{ij} := k(X_t^{(i)}, X_t^{(j)})\mathbf{I}_d. \quad (32)$$

Here $\mathbf{I}_d$ is the $d$-dimensional identity matrix, $W$ is a $dn$-dimensional Wiener process and $k$ is a kernel function $\mathbb{R}^d \times \mathbb{R}^d \to \mathbb{R}$. In our numerical experiments we chose the Gaussian kernel $k(s, y) = \frac{1}{2}\alpha \exp\left(-\frac{\|x-y\|^2}{2\kappa^2}\right)$. The kernel parameters are chosen to be $\alpha = 0.3, \kappa = 0.5$ to ensure a strong correlation between a wide range of landmarks. Note that the diffusion coefficient $Q$ is state-dependent. To demonstrate the necessity of constructing such a state-dependent diffusion coefficient, consider the process $\tilde{X}$ solving the SDE

$$\mathrm{d}\tilde{X}_t = Q(v)\mathrm{d}W_t, \quad (33)$$

where the diffusion constant is constant. As a result, the system becomes linear. The processes defined by Equation (32) and Equation (33) are fundamentally different. In particular, Equation (32) guarantees that $t \mapsto X_t$ generates a stochastic flow of diffeomorphisms (Sommer et al., 2021). This diffeomorphic setting preserves the shape structure during evolution, whereas the linear process defined by Equation (33) does not. As illustrated in Figure 13, a visual comparison (using identical driving Wiener processes) reveals that the linear process disrupts the shape topology, leading to overlaps and intersections. The diffusion process defined by Equation (33) can however be used as auxiliary process in the construction of guided proposals. We opted for this choice in our numerical experiments.

In our numerical experiments, we chose one ellipse as the starting point and another ellipse as the endpoint of the bridge. Each ellipse is discretized by 50 landmarks, leading to the dimension of $X_t$ being 100. We took $T = 1.0$. In the leftmost column of Figure 14 we fix a Wiener process and show on the top- and middle row the guided proposal and neural bridge using this Wiener process. We observed that due to the very simple choice of auxiliary process, the guided proposal has difficulty reaching the final state. In fact, we had to increase $\epsilon^2$ to $2e - 3$ for not running into numerical instabilities. Here, one can see that the additional learning by the neural bridge gives much better performance. In the bottom row, we used the guided proposal with the same Wiener process as initialization, augmented by running

5000 pCN iterations and plotted the final iteration. From this, one can see that these iterations provide another way to improve upon the guided proposal in the top row. The other columns repeat the same experiment with different Wiener process initializations. Due to the high-dimension of the problem, repeated simulation required for pCN steps may be computationally expensive.

We benchmarked our method on the four test cases described above, comparing it to two other deep learning-based approaches in Table 1. The comparison considers the number of network parameters and total training time, with all methods trained using the same number of gradient descent steps and batch sizes. Our method outperforms both alternatives in terms of model size and training efficiency.

## 6. Conclusions and Limitations

We propose the neural guided diffusion bridge, a novel method for simulating diffusion bridges that enhances guided proposals through variational inference, eliminating the need for MCMC or SMC. This approach enables efficient independent sampling with comparable quality in challenging tasks where existing score-learning-based methods struggle. Extensive experiments, including both quantitative and qualitative evaluations, validate the effectiveness of our method. However, as the framework is formulated variationally and optimized by minimizing $\mathrm{D}_{\mathrm{KL}}(\mathbb{P}_\theta^\bullet \| \mathbb{P}^\star)$, it exhibits mode-seeking behaviour, potentially limiting its ability to explore all modes compared to running multiple MCMC chains. Despite this limitation, our method provides a computationally efficient alternative to guided proposals, particularly in generating independent samples from the conditioned process.

Our approach focuses on better approximating the drift of the conditioned process while keeping the guiding term that ensures the process hits $v$ at time $T$ relatively simple. In future work, an interesting direction to obtain improved results consists in trying to jointly learn $\vartheta_\theta$ and the parameters of the linear process ($\tilde{\sigma}, B, \beta$). Also, as the gradient updating relies on backpropagating through the whole numerical SDE solvers, techniques such as the stochastic adjoint sensitivity method (Li et al., 2020) and adjoint matching (Domingo-Enrich et al., 2025) can be introduced to improve the efficiency of the computation. Another venue of future research consists of extending our approach to conditioning on partial observations at multiple future times.

## Acknowledgements

The authors thank anonymous reviewers and Thorben Pieper-Sethmacher (TU Delft) for their valuable comments and suggestions, which have helped to improve the clarity and quality of this work. The work presented in this paper was supported by the Villum Foundation Grant 40582, and the Novo Nordisk Foundation grant NNF18OC0052000.

## Impact statement

This paper presents work whose goal is to advance the field of Machine Learning and Statistics. There are many potential societal consequences of our work, none of which we feel must be specifically highlighted here.

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

# A. Theoretical Details

## A.1. Preconditioned Crank-Nicolson

Algorithm 2 is a special case of Algorithm 4.1 of (Mider et al., 2021).

---

**Algorithm 2** Preconditioned Crank-Nicolson scheme for guided proposals

---

1: **Input:** Discrete time grid $\mathcal{T} := \{t_m\}_{m=0,1,\dots,M}$, tuning parameter $\eta \in [0,1)$, number of required samples $K$
2: Solve Equation (15) on $\mathcal{T}$, obtain $\{\tilde{H}(t_m)\}, \{\tilde{r}(t_m, \cdot)\}$ using Equations (13) and (14).
3: Sample $w = \{w_{t_m}\}$ on $\mathcal{T}$.
4: Solve Equation (7) on $\mathcal{T}$ with $w = \{w_{t_m}\}$, obtain $y = \{y_{t_m}\}$.
5: **repeat**
6:     Sample new innovations $z = \{z_{t_m}\}$ on $\mathcal{T}$ independently.
7:     Set $w^\circ = \eta w + \sqrt{1 - \eta^2} z$.
8:     Solve Equation (7) on $\mathcal{T}$ with $z = \{z_{t_m}\}$, obtain $y^\circ = \{y^\circ_{t_m}\}$.
9:     Compute $A = \Psi(y^\circ)/\Psi(y)$ with $\{y^\circ_{t_m}\}$ and $\{y_{t_m}\}$ using Equation (11).
10:    Draw $U \sim \mathcal{U}(0,1)$.
11:    **if** $U < A$ **then**
12:       $y \leftarrow y^\circ$ and $w \leftarrow w^\circ$
13:    **end if**
14:    Save $y$.
15: **until** Sample counts $> K$.

---

## A.2. Proof of Theorem 4.3

*Proof.* Consider the KL divergence between $\mathbb{P}^\bullet_\theta$ and $\mathbb{P}^\star$:

$$D_{\mathrm{KL}}(\mathbb{P}^\bullet_\theta \| \mathbb{P}^\star) = \mathbb{E}^\bullet \left[ \log \left( \frac{d\mathbb{L}^\bullet_\theta}{d\mathbb{L}^\star} \right)(X) \right] = \mathbb{E}^\bullet \left[ \log \left( \frac{d\mathbb{L}^\bullet_\theta}{d\mathbb{L}^\circ} \cdot \frac{d\mathbb{L}^\circ}{d\mathbb{L}^\star} \right)(X) \right]$$

$$= \mathbb{E}^\bullet \left[ \log \left( \frac{d\mathbb{L}^\bullet_\theta}{d\mathbb{L}^\circ}(X) \right) \right] - \mathbb{E}^\bullet \left[ \log \left( \frac{d\mathbb{L}^\star}{d\mathbb{L}^\circ}(X) \right) \right]. \tag{34a}$$

By Girsanov's theorem,

$$\mathbb{E}^\bullet \left[ \log \left( \frac{d\mathbb{L}^\bullet_\theta}{d\mathbb{L}^\circ}(X) \right) \right] = \mathbb{E}^\bullet \left[ \log \frac{d\mathbb{P}^\bullet_\theta}{d\mathbb{P}^\circ} \right] \tag{35a}$$

$$= \mathbb{E}^\bullet \left[ \int_0^T \vartheta_\theta(t, X_t) dW^\circ_t - \frac{1}{2} \int_0^T \|\vartheta_\theta(t, X_t)\|^2 dt \right]$$

$$= \mathbb{E}^\bullet \left[ \int_0^T \vartheta_\theta(t, X_t) dW^\bullet_t + \frac{1}{2} \int_0^T \|\vartheta_\theta(t, X_t)\|^2 dt \right]$$

$$= \mathbb{E}^\bullet \left[ \frac{1}{2} \int_0^T \|\vartheta_\theta(t, X_t)\|^2 dt \right], \tag{35b}$$

where the stochastic integral vanishes because of the martingale property of the Itô integral. The first equality follows from Equation (18). By Equation (8)

$$\mathbb{E}^\bullet \left[ \log \left( \frac{d\mathbb{L}^\star}{d\mathbb{L}^\circ}(X) \right) \right] = \mathbb{E}^\bullet \left[ \int_0^T G(t, X_t) dt \right] + \log \frac{\tilde{h}(0, x_0)}{h(0, x_0)}. \tag{36}$$

Substituting Equation (35b) and Equation (36) into Equation (34a) gives

$$D_{\mathrm{KL}}(\mathbb{P}^\bullet_\theta \| \mathbb{P}^\star) = \mathbb{E}^\bullet \int_0^T \left\{ \frac{1}{2} \|\vartheta_\theta(t, X_t)\|^2 - G(t, X_t) \right\} dt - \log \frac{\tilde{h}(0, x_0)}{h(0, x_0)} = L(\theta) - \log \frac{\tilde{h}(0, x_0)}{h(0, x_0)} \geq 0, \tag{37}$$

with $L(\theta)$ as defined in Equation (20). $\qquad\square$

## A.3. SDE Gradients

We now derive the gradient of Equation (23) with respect to $\theta$, on a fixed Wiener realization $w^{\bullet(n)} = \{w_{t_m}^{\bullet(n)}\}$. As discussed, $x_{t_m}^{\bullet(n)} = \phi_\theta(w_{t_m}^{\bullet(n)})$ is implemented as a numerical SDE solver $f_\theta(w_{t_m}^{\bullet(n)}, t_{m-1}, x_{t_{m-1}}^{\bullet(n)}), m \geq 1$ that takes the previous step $(t_{m-1}, x_{t_{m-1}}^{\bullet(n)})$ as additional arguments. As $x_{t_{m-1}}^{\bullet(n)}$ also depends on $\theta$, the gradient with respect to $\theta$ needs to be computed recursively. Specifically, with $x_{t_m}^{\bullet(n)} = f_{\theta,m} = f_\theta(w_{t_m}^{\bullet(n)}, t_{m-1}, x_{t_{m-1}}^{\bullet(n)})$

$$\nabla_\theta \left( \frac{1}{2} \|\vartheta_\theta(t_{m-1}, \phi_\theta(w_{t_{m-1}}^{\bullet(n)}))\|_2^2 \right) = \nabla_\theta \left( \frac{1}{2} \|\vartheta_\theta(t_{m-1}, f_{\theta,m-1})\|_2^2 \right) \tag{38a}$$

$$= [\nabla_\theta \vartheta_\theta(t_{m-1}, f_{\theta,m-1}))]^T \vartheta_\theta(t_{m-1}, f_{\theta,m-1}) \tag{38b}$$

$$= \left[ \frac{\partial \vartheta_\theta(t_{m-1}, f_{\theta,m-1})}{\partial \theta} + \frac{\partial \vartheta_\theta(t_{m-1}, f_{\theta,m-1})}{\partial f_{\theta,m-1}} \cdot \nabla_\theta f_{\theta,m-1} \right]^T \vartheta_\theta(t_{m-1}, f_{\theta,m-1}) \tag{38c}$$

$$= \left[ \frac{\partial \vartheta_\theta(t_{m-1}, f_{\theta,m-1})}{\partial \theta} + \frac{\partial \vartheta_\theta(t_{m-1}, f_{\theta,m-1})}{\partial f_{\theta,m-1}} \cdot \left( \frac{\partial f_{\theta,m-1}}{\partial \theta} + \frac{\partial f_{\theta,m-1}}{\partial f_{\theta,m-2}} \cdot \nabla_\theta f_{\theta,m-2} \right) \right]^T \vartheta_\theta(t_{m-1}, f_{\theta,m-1}) \tag{38d}$$

$$= \left[ \frac{\partial \vartheta_\theta(t_{m-1}, f_{\theta,m-1})}{\partial \theta} + \frac{\partial \vartheta_\theta(t_{m-1}, f_{\theta,m-1})}{\partial f_{\theta,m-1}} \cdot \left( \frac{\partial f_{\theta,m-1}}{\partial \theta} + \sum_{i=1}^{m-2} \left( \prod_{j=i+1}^{m-1} \frac{\partial f_{\theta,j}}{\partial f_{\theta,j-1}} \right) \frac{\partial f_{\theta,i}}{\partial \theta} \right) \right]^T \vartheta_\theta(t_{m-1}, f_{\theta,m-1}), \tag{38e}$$

Similarly, the gradient of $G$ with respect to $\theta$ can also be computed recusively:

$$\nabla_\theta G(t_{m-1}, \phi_\theta(w_{t_{m-1}}^{\bullet(n)})) = \frac{\partial G(t_{m-1}, f_{\theta,m-1})}{\partial f_{\theta,m-1}} \cdot \left( \frac{\partial f_{\theta,m-1}}{\partial \theta} + \sum_{i=1}^{m-2} \left( \prod_{j=i+1}^{m-1} \frac{\partial f_{\theta,j}}{\partial f_{\theta,j-1}} \right) \frac{\partial f_{\theta,i}}{\partial \theta} \right). \tag{39}$$

The gradient of $L(\theta)$ can be approximated by:

$$\nabla_\theta L(\theta) \approx \frac{1}{N} \sum_{n=1}^{N} \sum_{m=1}^{M} \left\{ \nabla_\theta \left( \frac{1}{2} \|\vartheta_\theta(t_{m-1}, \phi_\theta(w_{t_{m-1}}^{\bullet(n)}))\|_2^2 \right) - \nabla_\theta G(t_{m-1}, \phi_\theta(w_{t_{m-1}}^{\bullet(n)})) \right\} \delta t. \tag{40}$$

The realization of $f_\theta$ depends on the chosen numerical integrator. We choose Euler-Maruyama as the integrator used for all the experiments conducted in Section 5. Under this scheme, $f_\theta$ is:

$$f_\theta(w_{t_m}^{\bullet(n)}, t_{m-1}, x_{t_{m-1}}^{\bullet(n)}) = x_{t_{m-1}}^{\bullet(n)} + (b + a\tilde{r} + \sigma\vartheta_\theta)(t_{m-1}, x_{t_{m-1}}^{\bullet(n)}) + \sigma(t_{m-1}, x_{t_{m-1}}^{\bullet(n)})w_{t_m}^{\bullet(n)}, \tag{41}$$

with $w_{t_m}^{\bullet(n)} \sim \mathcal{N}(0, (t_m - t_{m-1})\mathbf{I}_d)$. The derivatives can be computed accordingly:

$$\frac{\partial f_{\theta,m}}{\partial \theta} = \sigma(t_{m-1}, x_{t_{m-1}}^{\bullet(n)}) \frac{\partial \vartheta_\theta(t_{m-1}, x_{t_{m-1}}^{\bullet(n)})}{\partial \theta} \tag{42a}$$

$$\frac{\partial f_{\theta,m}}{\partial f_{\theta,m-1}} = 1 + \frac{\partial (b + a\tilde{r} + \sigma\vartheta_\theta)}{\partial x_{t_{m-1}}^{\bullet(n)}}(t_{m-1}, x_{t_{m-1}}^{\bullet(n)}) + \frac{\partial \sigma}{\partial x_{t_{m-1}}^{\bullet(n)}}(t_{m-1}, x_{t_{m-1}}^{\bullet(n)})w_{t_m}^{\bullet(n)}. \tag{42b}$$

The automatic differentiation can save all the intermediate Equation (42a) and Equation (42b), which enables to compute $\nabla_\theta L(\theta)$.

## B. Experiment Details

### B.1. Code Implementation

The codebase for reproducing all the experiments conducted in the paper is available in https://github.com/bookdiver/neuralbridge

## B.2. Linear Processes

**Brownian bridges:** If $dX_t = \gamma dt + \sigma dW_t$, then

$$\log h(t, x) = \log p(T, v \mid t, x) = -\frac{1}{2}\log(2\pi\sigma^2(T-t)) - \frac{(v - x - \gamma(T-t))^2}{2\sigma^2(T-t)}. \tag{43}$$

If $d\tilde{X}_t = \sigma dW_t$, then $\log \tilde{h}(t, x)$ is obtained by taking $\gamma = 0$ in the preceding display.

Therefore, in this case we can compute the lower bound on the loss: $L(\theta) \geq \log \frac{\tilde{h}(0,x_0)}{h(0,x_0)} = \frac{(v-x-\gamma T)^2 - (v-x)^2}{2\sigma^2 T}$. Moreover, the optimal map $\vartheta_{\theta_{\mathrm{opt}}}$ is given by $\vartheta_{\theta_{\mathrm{opt}}}(t, x) = \sigma(\partial_x \log h(t, x) - \partial_x \log \tilde{h}(t, x)) = -\frac{\gamma}{\sigma}$.

In the numerical experiment, we took $\epsilon = 10^{-5}$. The map $\vartheta_\theta$ is modeled by a fully connected neural network with 3 hidden layers and 20 hidden dimensions for each layer. The model is trained with 25,000 independently sampled full trajectories of $X^\bullet$. The batch size was taken to be $N = 50$ and the time step size $\delta t = 0.002$, leading to in total $M = 500$ time steps. The network was trained using the Adam (Kingma & Ba, 2017) optimizer with learning rate 0.001.

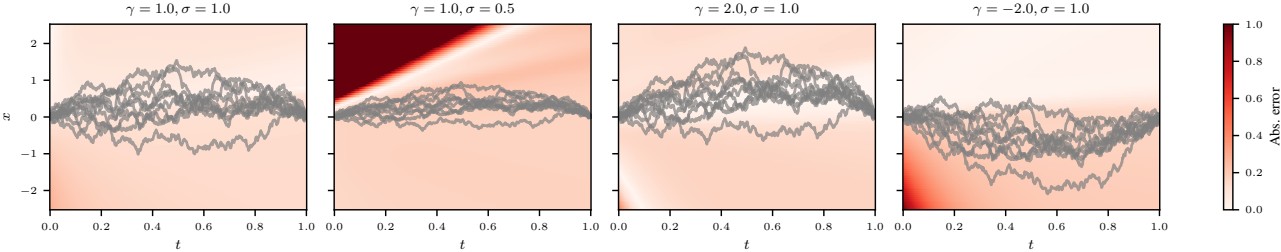

*Figure 2.* Evaluations of trained neural networks $\vartheta_\theta(t, x)$ against the optimal maps $\vartheta_{\theta_{\mathrm{opt}}}(t, x)$ under different settings of $\gamma, \sigma$, where the background colour intensities indicate the absolute error $|\vartheta_\theta - \vartheta_{\theta_{\mathrm{opt}}}|$. In each panel, 10 independent samples from the guided proposal are shown in grey to indicate the sampling regions where one expects the error to be smallest.

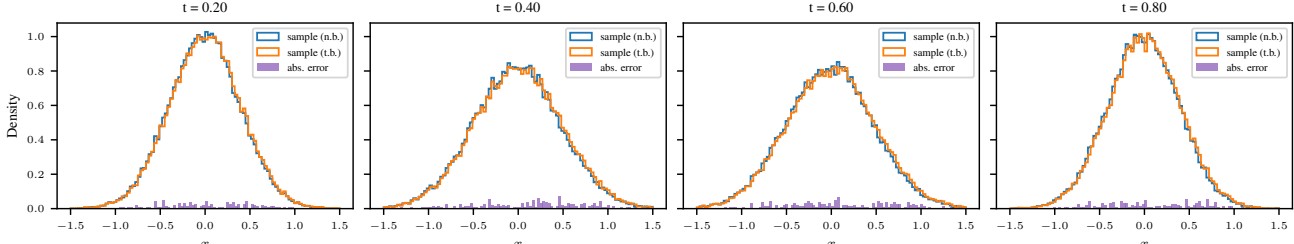

*Figure 3.* Comparison of marginal distributions at different time slices of the learned neural bridge (n.b.) and analytical true Brownian bridge (t.b.) under the setting $\gamma = 1.0, \sigma = 1.0$, conditioned on $v = 0.0$. The purple bars show the absolute error (abs. error) of each bin between the learned neural bridge and the histograms obtained by forward sampling from the true bridge process, which are expected to be low when two distributions are close in terms of their shapes. The histograms are made from 50,000 independent trajectory samples.

**Ornstein-Uhlenbeck bridge:** When conditioning Equation (26) on $v$, the conditioned process $X^\star$ satisfies the SDE

$$dX_t^\star = \left\{ \gamma(\mu - X_t^\star) + \frac{2\gamma e^{-\gamma(T-t)}}{1 - e^{-2\gamma(T-t)}} \left[(v - \mu) - e^{-\gamma(T-t)}(X_t^\star - \mu)\right] \right\} dt + \sigma dW_t, \tag{44}$$

which is obtained from the transition density given by Equation (26) is:

$$p(T, y \mid t, x) = \frac{1}{\sqrt{2\pi \Sigma_{t,T}^2}} \exp\left(-\frac{(v - m_{t,T}(x))^2}{2\Sigma_{t,T}^2}\right), \tag{45a}$$

$$m_{s,t}(x) = \mu + (x - \mu)e^{-\gamma(t-s)}, \tag{45b}$$

$$\Sigma_{s,t}^2 = \frac{\sigma^2}{2\gamma}\left(1 - e^{-2\gamma(t-s)}\right). \tag{45c}$$

Since the auxiliary process is chosen the same as the Brownian case, one can easily show the optimal value of $\vartheta_\theta(t, x)$ to be

$$\vartheta_{\theta_{\text{opt}}}(t, x) = \frac{2\gamma e^{-\gamma(T-t)}}{\sigma(1 - e^{-2\gamma(T-t)})}\left[(v - \mu) - e^{-\gamma(T-t)}(x - \mu)\right] + \frac{(v - x)}{\sigma(T - t)}. \tag{46}$$

Therefore, the lower bound on $\theta \mapsto L(\theta)$ is given by

$$\log \frac{\tilde{h}(0, x_0)}{h(0, x_0)} = -\frac{1}{2}\log(2\pi\sigma^2(T - t)) - \frac{(v - x)^2}{2\sigma^2(T - t)} + \frac{1}{2}\log(2\pi\Sigma_{t,T}^2) + \frac{(v - m_{t,T}(x))^2}{2\Sigma_{t,T}^2}. \tag{47}$$

We repeated the same numerical and experimental settings as the previous Brownian example, except for training with 50,000 samples to obtain better results.

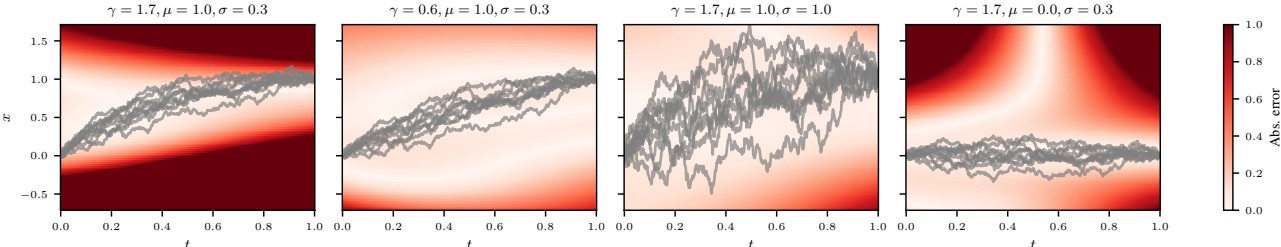

*Figure 4.* Evaluations of trained neural networks $\vartheta_\theta(t, x)$ against the optimal maps $\vartheta_{\theta_{\text{opt}}}(t, x)$ under different settings of $\gamma, \mu, \sigma$, where the background colour intensities indicate the absolute error $|\vartheta_\theta - \vartheta_{\theta_{\text{opt}}}|$. In each panel, 10 independent samples from the guided proposal are shown in grey to indicate the sampling regions where one expects the error to be smallest. Except the rightmost setting with $\gamma = 1.7, \mu = 0.0, \sigma = 0.3$, all the processes are conditioned on $v = 1.0$, whereas in the rightmost case, the process is conditioned on $v = 0.0$.

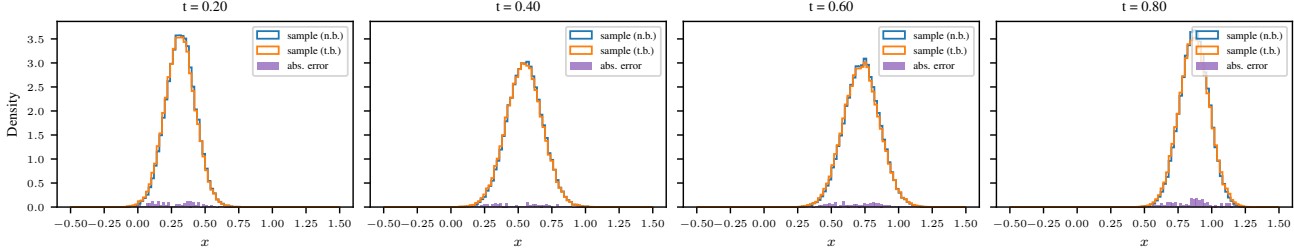

*Figure 5.* Comparison of marginal distributions at different time slices of the learned neural bridge (n.b.) and analytical true OU bridge (t.b.) under the setting $\gamma = 1.7, \mu = 1.0, \sigma = 0.3$, conditioned on $v = 1.0$. The purple bars show the absolute error (abs. error) of each bin. The histograms are made from 50,000 independent trajectory samples.

## B.3. Cell Diffusion Process

For the benchmark tests, we adapt the published guided proposal implementations of the corresponding methods to fit into our test framework with possibly minor modifications. Specifically, the original guided proposal codebase is implemented with

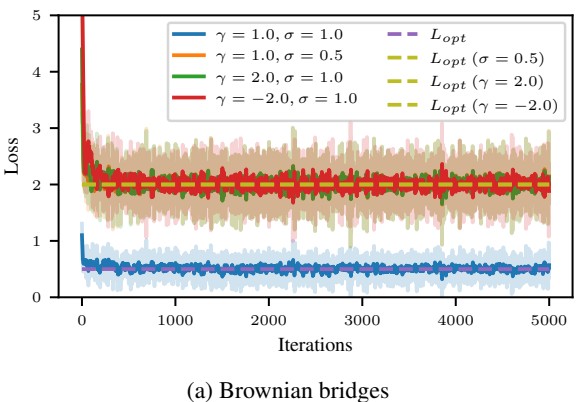 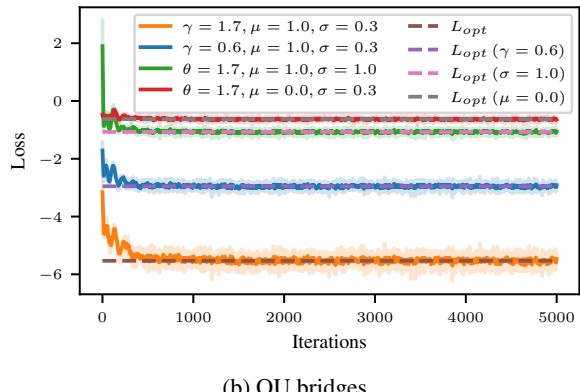

(a) Brownian bridges          (b) OU bridges

*Figure 6.* Loss curves of training Brownian and OU bridges. For Brownian bridges, the SDE parameters are taken as $\gamma = 1.0, \sigma = 1.0$, for the OU bridges, the parameters are taken as $\gamma = 1.7, \mu = 1.0, \sigma = 0.3$ unless any of them is specified.

Julia in [1], we rewrite it in JAX (Bradbury et al., 2018); the score matching bridge repository is published in [2]. additionally, as also reported by the authors, (Heng et al., 2022) introduces two score-matching-based bridge simulation schemes, reversed and forward simulation, and the forward simulation relies on the reversed simulation, and learning from approximated reversed bridge can magnifies the errors due to progressive accumulation. Therefore, we only compare our method with the reversed bridge earning to avoid error accumulations; the adjoint bridge is implemented in [3].

**Normal event:** We took $\epsilon^2 = 10^{-10}$ and $\vartheta_\theta$ is modeled as a fully-connected network with 3 hidden layers and 32 hidden dimensions per layer, activated by LipSwish. We trained the model for 5,000 gradient descent updating iterations. In each step, we sampled a batch of 100 independent trajectories of $X^\bullet$ under the current $\theta$. The numerical sampling time step size is $\delta t = 0.01$. Therefore, in total $M = 400$ discrete steps of a single trajectory. The Adam optimizer with learning rate of $1e - 3$ was used for optimization. For the guided proposal, we set $\eta = 0.98$, and ran one MCMC chain for 10,000 iterations. This resulted in an acceptance percentage of $21.38\%$. The initial $5,000$ iterations were considered as burn-in samples. After burn-in, we collected the samples and subsampled them by taking every 133 samples to obtain 30 samples. In the following experiments, if not explicitly stated, all the samples from the guided proposal are similarly obtained from one MCMC chain by subsampling from the outputs. For the score matching and adjoint forward methods, we deployed the given network structures provided in their codebases. As a reference, we sampled from the forward process until we obtained 30 samples satisfying the inequality $\|LX_T - v\| \le 0.01$. These samples are shown in grey.

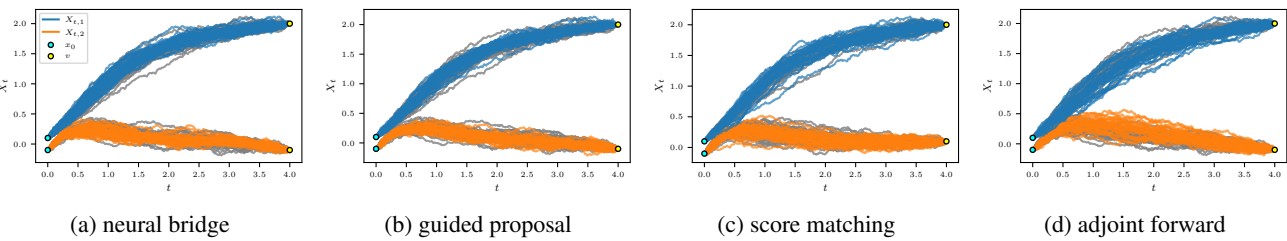

(a) neural bridge      (b) guided proposal      (c) score matching      (d) adjoint forward

*Figure 7.* Visualization of 30 simulated bridge trajectories under normal event conditioning, using various sampling methods. All samples are independently drawn, except those generated by the guided proposal. As a reference, in grey, we added trajectories from the unconditional forward process, with samples that satisfy the condition $\|LX_T - v\| > 0.01$ rejected.

**Rare event:** The setups for conditioning on rare events of the neural guided bridge are replicated from the previous normal event case, except for the MCMC is running with sightly increased tuning parameter $\eta = 0.99$ and for 20,000 iterations and the first 10,000 iterations is discarded as the burn-in period. The acceptance percentage is $22.29\%$. For the score matching,

---

[1]https://juliapackages.com/p/bridge
[2]https://github.com/jeremyhengjm/DiffusionBridge
[3]https://github.com/libbylbaker/forward_bridge

since we only use the reversed bridge, where learning is independent of the event we condition on , we directly deploy the trained score approximation from the previous case. For the adjoint forward, we fix the neural network architecture and training scheme, changing only the conditioned target.

As a reference, we sampled the forward process $100,000$ times. This time however, none of the samples ended near the point we condition on, confirming we are dealing with a rare event.

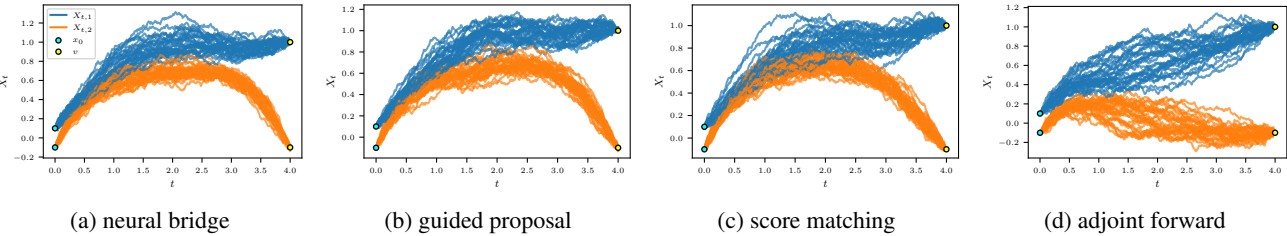

| (a) neural bridge | (b) guided proposal | (c) score matching | (d) adjoint forward |

*Figure 8.* Visualization of 30 simulated bridge trajectories under rare event conditioning, using various sampling methods. All samples are independently drawn, except those generated by the guided proposal. Contrary to Figure 7, no reference trajectories were added as extensive forward simulation of the process did not yield any samples satisfying $\|LX_t - v\| < 0.01$.

**Multi-modality:** All neural network architectures and training settings match those used in previous examples. We set $\delta t = 0.01$ and $M = 500$. For the guided proposal, a single chain is run for $50,000$ iterations with $\eta = 0.9$, discarding the first $20,000$ as burn-in. The acceptance percentage is $26.81\%$. As in the rare event case, no valid samples were found from forward simulating $100,000$ times the (unconditioned) forward process.

Figure 10 shows marginal distributions for both unconditioned and conditioned processes, using the guided proposal and the neural bridge. At $t = 3.0$ and $t = 4.0$, multiple peaks appear in the marginal densities, suggesting multi-modality. This is consistent with the unconditioned sampling, where multiple modes are also visible. However, neither score matching nor the adjoint forward method captures these modes. In contrast, the neural bridge and guided proposal yield similar marginals that accurately reflect the multi-modal nature of the process.

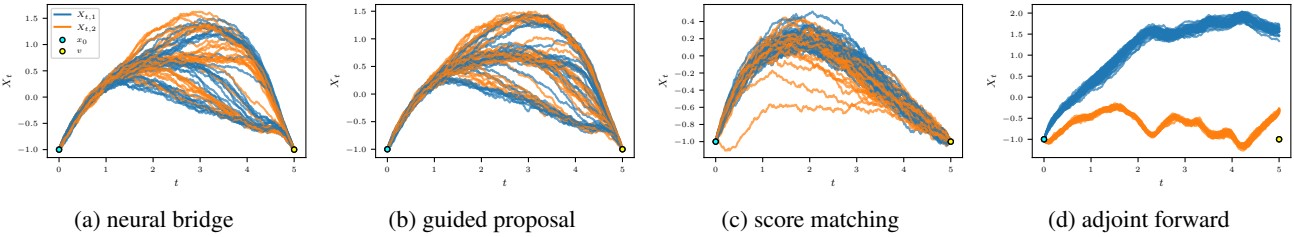

| (a) neural bridge | (b) guided proposal | (c) score matching | (d) adjoint forward |

*Figure 9.* Visualization of 30 simulated bridge trajectories under multi-modal event conditioning, using various sampling methods. All samples are independently drawn, except those corresponding to the guided proposal.

### B.4. FitzHugh-Nagumo Model

**Normal event:** We set $\delta t = 0.005$, which leads to $M = 400$ time steps. We took $\epsilon^2 = 1e-8$. For the neural guided bridge, $\vartheta_\theta(t, x)$ is constructed as a fully connected neural network with 4 hidden layers and 32 hidden dimensions at each layer, activated by LipSwish functions. The training is done for $5,000$ gradient descent steps. In each step, we generated $N = 100$ independent samples from the current $X^\bullet$ for Monte Carlo estimation. The network is optimized by Adam with a learning rate of $1e-3$. For the guided proposal, we set $\eta = 0$ as suggested in (Bierkens et al., 2020) and ran one chain for $50,000$ iterations with a burn-in of $20,000$ steps, obtaining $64.41\%$ acceptance percentage. The reference is obtained by sampling the (unconditional) forward process filtering samples with $\|LX_T - v\| \leq 0.01$, as $v = \begin{bmatrix} 1.0 \end{bmatrix}$ is an event around which the process is likely to reach a small ball, one can expect to easily obtain sufficient samples that meets the filtering condition, it turns out that we only need to sample the forward process for $10,000$ to obtain $450$ vaild samples, and we only show the first 30 in the figure.

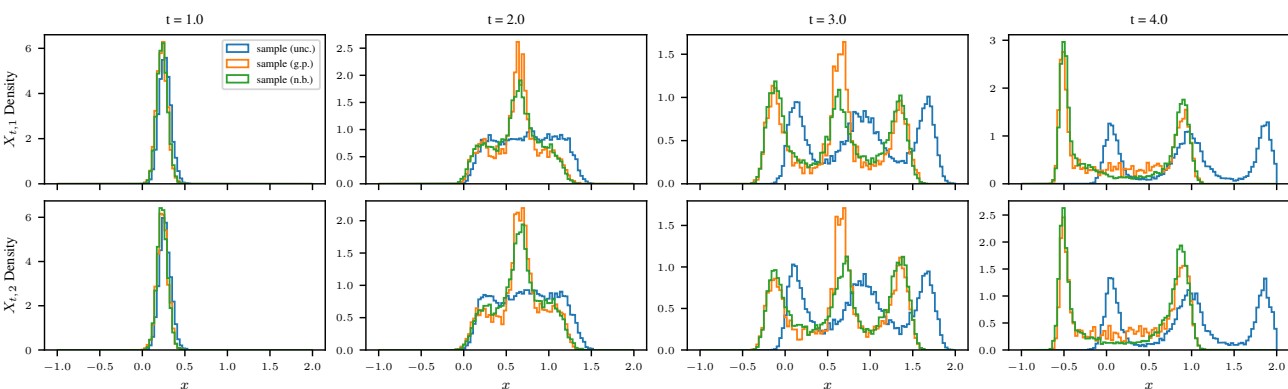

*Figure 10.* Marginal distributions of $X$ at selected time points for three settings: the unconditioned process (unc.), the neural bridge (n.b.), and the guided proposal sampled with pCN (g.p.). All trajectories start at $x_0 = \begin{bmatrix} -1 & -1 \end{bmatrix}^\top$ and, The neural bridge and the guided proposal are conditioned to reach $v = \begin{bmatrix} -1 & -1 \end{bmatrix}^\top$ at the terminal time $T = 5$. Each histogram is based on 10,000 samples—independent draws for unc. and n.b. and subsampled MCMC draws for g.p. The first and second components of X are displayed in the top and bottom rows, respectively.

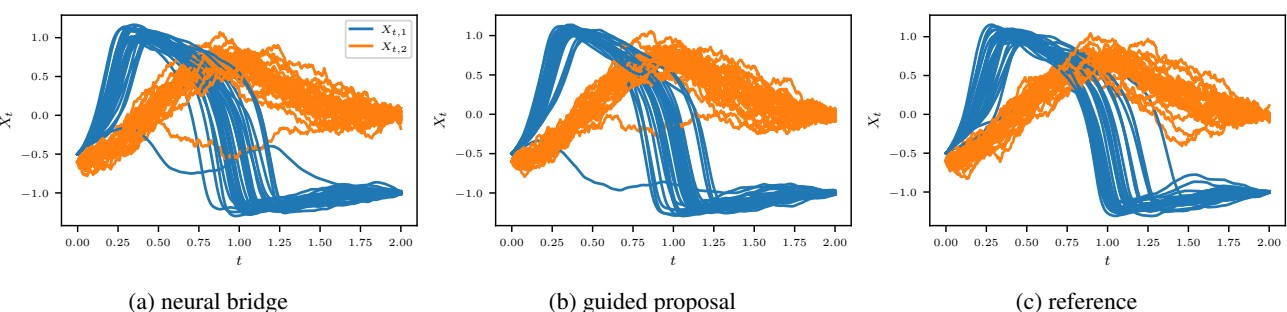

      (a) neural bridge               (b) guided proposal               (c) reference

*Figure 11.* Visualization of 30 simulated bridge trajectories under normal event conditioning from the learned neural bridge and pCN sampling of the guided proposal. The reference trajectories in (c) are obtained by forward sampling the unconditioned process, keeping only samples that satisfy the condition $\|LX_T - v\| \leq 0.01$.

**Rare event:** We duplicate the neural network training setting as before, including the used network structure and training hyperparameters. For the guided proposal we set the pCN-parameter to $\eta = 0.9$ and ran one chain for 50,000 iterations which yielded a Metropolis-Hastings acceptance percentage equal to 22.46%. The initial 20,000 iterations are considered burnin samples. As a reference, we forward sampled 200,000 paths of the unconditioned process. Of those, 35 samples satisfied $\|LX_t - v\| < 0.01$, which is about 0.02% of the samples. As expected, in the "normal event" case, this percentage was higher (4.5%).

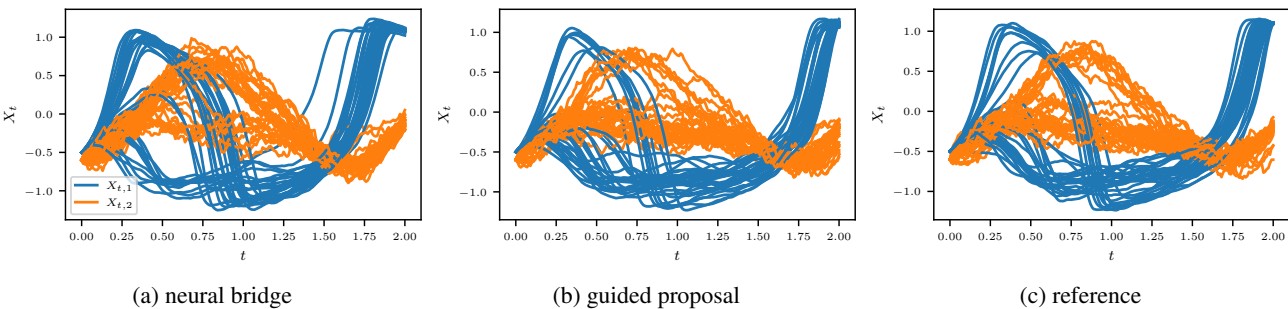

(a) neural bridge        (b) guided proposal        (c) reference

*Figure 12.* Visualization of 30 simulated bridge trajectories under rare event conditioning from the learned neural bridge and pCN sampling of the guided proposal. The reference trajectories in (c) are obtained by forward sampling the unconditioned process, and filtered the results by the condition $\|LX_T - v\| \leq 0.01$.

## B.5. Stochastic Landmark Matching

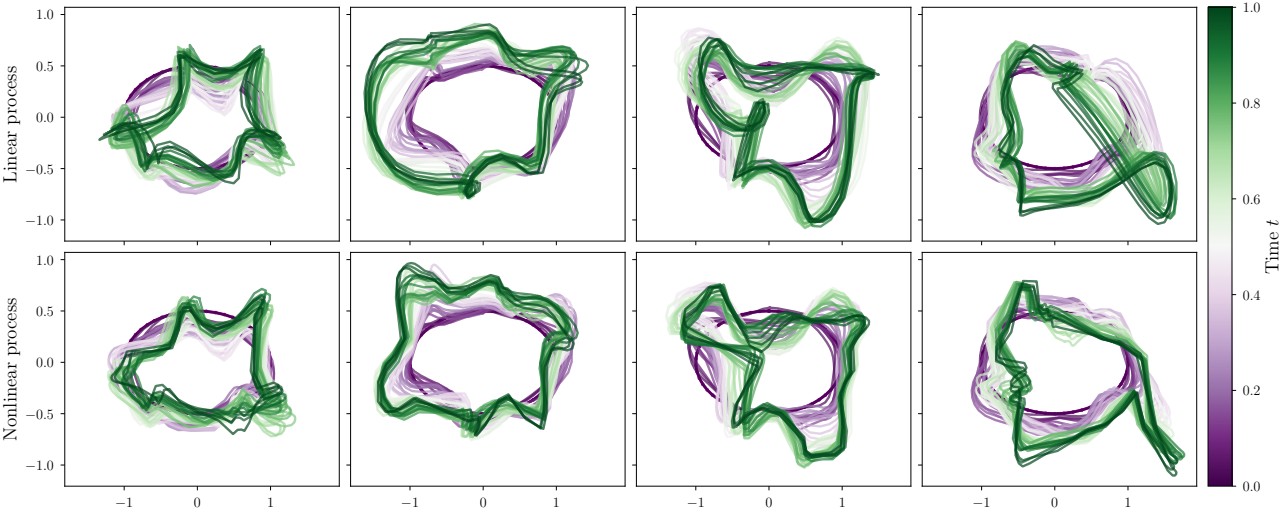

*Figure 13.* Comparison of unconditioned samples from the linear process (Equation (33)) and the nonlinear process (Equation (32)). Top row: 4 independent samples from Equation (33). Bottom row: corresponding samples from Equation (32), each using the same Wiener process realization as in the top row.

The observation noise variance is set as $\epsilon^2 = 2e - 3$, as we find too small values of $\epsilon$ will cause numerical instability. We deploy the neural network architecture suggested in (Heng et al., 2022) to model $\vartheta_\theta$, whose encoding part is a two-layer MLP with 128 hidden units at each layer, and the decoding part is a three-layer MLP with hidden units of 256, 256, and 128 individually. The network is activated by tanh, and trained with 240,000 independent samples from $X^\bullet$ with batch size $N = 8$, optimized by Adam with an initial learning rate of $7.0e - 4$ and a cosine decay scheduler. For the guided proposal, we run 4 chain for 5,000 iterations each and drop the first 1,000 iterations as the burn-in, with $\eta = 0.95$ and obtain 12.62% acceptance percentage on average over the 4 chains.

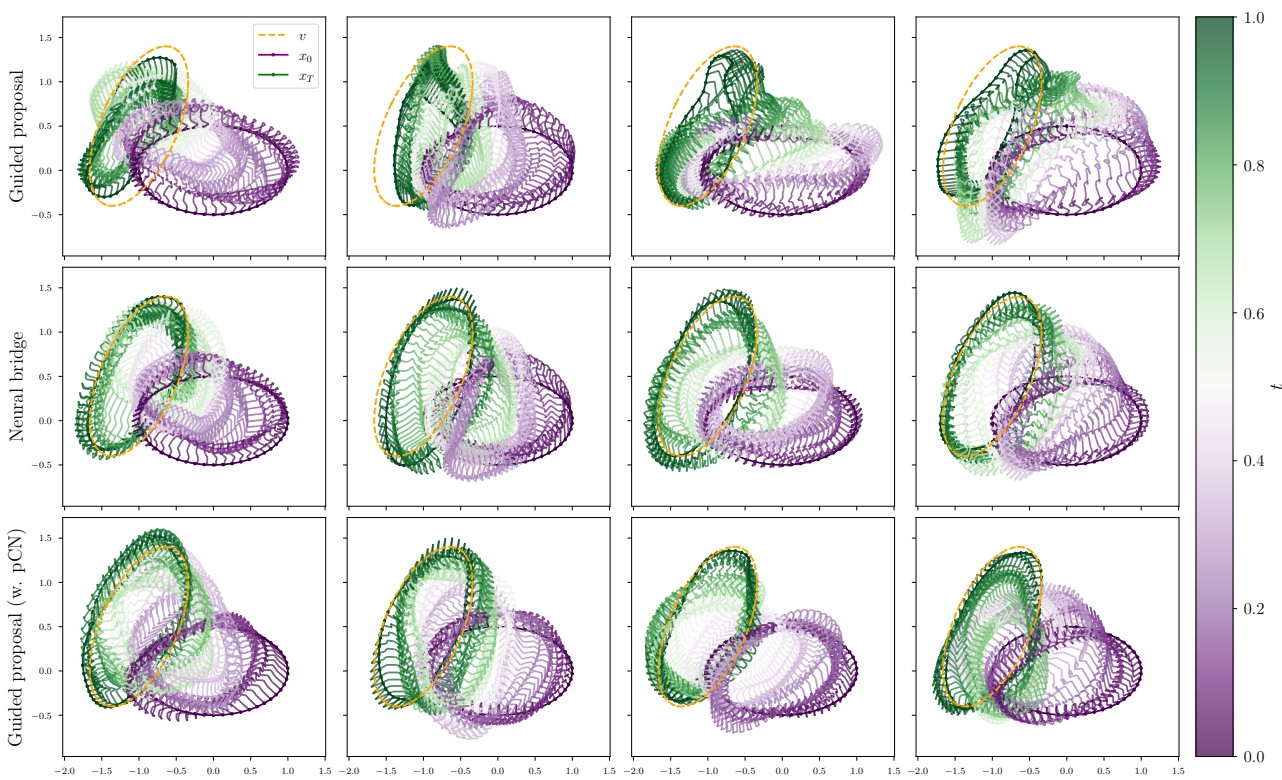

*Figure 14.* 4 realizations of sampling from the guided proposal, the trained neural bridge and the final iteration after updating the guided proposal with $5,000$ pCN-steps. Top row: 4 samples of the guided proposal using 4 independent Wiener process realisations; middle row: 4 samples from the trained neural bridge using the same Wiener realisations as the top row; Bottom row: the final outputs of 4 independent chains updating the guided propsals using pCN-steps. The chains are initialised with the same Wiener realisations as the top row.

