# OpenReview forum: "Neural Guided Diffusion Bridges"
_ICML.cc/2025/Conference — ICML 2025 poster_

### Official Review · Reviewer_XLMC · 2025-03-10

**Overall Recommendation:** 3

**Summary:**

The paper considers the challenging and widely-applicable problem of conditioning a reference diffusion process to sample rare events or desired outcomes.   Building on "guided proposal" approaches which construct the conditioned process for a cleverly-chosen tractable process, the authors propose to learn an additional control drift, parameterized using neural networks and minimizing a stochastic optimal control or mode-seeking KL objective via backpropagation through trajectories and the reparameterization trick.

The authors demonstrate the efficacy of the proposed approach on simple linear examples, along with cell dynamics, a FitzHugh-Nagumo excitable system, and a stochastic landmark matching problem.

**Claims And Evidence:**

The method is well-justified and incorporates useful conditioning information via the guided proposal.   As mentioned below, I encourage the authors to clarify the implementation and performance differences between the guided proposal and the proposed neural-network improvement, as the guided proposal provides a strong baseline in most of the experiments considered.

**Essential References Not Discussed:**

The authors might consider more fundamental citations regarding Prop 3.2, which appears to be well known.   If I am missing technical conditions relevant to the setting the authors have in mind (which require this recent result), then these might be stated.

*Non-essential recent work:*
- Denker et. al (NeurIPS 2024),  "DEFT: Efficient Fine-tuning of Diffusion Models by Learning the Generalised h-transform".   Of particular note  is the Stochastic control objective using Log Variance divergence (their conditional score matching training is outside the setting of this paper).
- Seong et. al (ICLR 2025):  "Transition Path Sampling with Improved Off-Policy Training of Diffusion Path Samplers":   context of Transition Path sampling in computational chemistry using a control objective and Log Variance divergence
- on the surface, Log Variance losses for stochastic control problems (i) leverage off-policy samples and (ii)  while this is an additional design choice, clever choice of exploration samples may mitigate mode-seeking behavior associated with the KL.   These considerations would be left for future work.

The authors might also be interested in recent work improving upon the adjoint method for backpropagation through trajectories (and references therein).
- Domingo-Enrich et. al 2024, "Adjoint Matching"

**Experimental Designs Or Analyses:**

The authors compare to existing methods such as score matching and adjoint bridges, along with the vanilla guided proposal.   While score matching and adjoint bridge appear to struggle with simple examples, the authors probe examples where the neural guided bridge improves over the vanilla guided proposal.

*Cell Dynamics*
- For cell dynamics with multi-modality in the conditional paths, the neural guided bridge appears to beter match the marginals of the unconditioned process in Fig 7.
-Is this the right evaluation?  (assuming the Original entry is  obtain from the unconditional process).   One could imagine that some modes of intermediate marginals do not result in the process hitting the desired v.

*FHN Model*
- the neural guided samples in Fig 9-10 are somewhat mode-seeking, but do not produce samples which interpolate between the modes (as in the guided proposal).   Here the reference process samples are appropriately filtered according to consistency with $v$.

**Methods And Evaluation Criteria:**

The authors consider a range of applications for validating the efficacy of the proposed method.


To ablate the necessity of the guided proposal, it might be interesting to consider the Neural Guided Bridge learning from the base drift directly (i.e. no guided proposal).  While this may fail in more complex settings, it would be interesting to evaluate for the Brownian Bridge and OU-Process in Sec. 5.1.

**Other Comments Or Suggestions:**

1) There is enough notation appearing that it might be useful to remind the reader of several aspects.
- In Eq 15, it would useful to be able to denote $\tilde{b}^{\circ}$ as a conditioned process with reference drift given by the linear $\tilde{b}_t$ in Eq 9, or simply rewrite the full form of Eq 15.
- In reading the paper, I had to remind myself that the Brownian bridge "prior information" $\frac{v-X_t^{\bullet}}{T-t}$ was not an ad-hoc decision, but rather is captured by $\tilde{b}^{\circ}$ (even for $\beta(t)=0, B(t) =0$ in choosing $\tilde{b}_t$).   It might be useful to state that the auxiliary process / guided proposal produces this term.



2) I had trouble parsing $q(v|y)$ as the likelihood at first glance, and it is written as $\ell(y|v) = q(v|y)$ in line 150L.   Please mention the desired role of $v$ before Prop 3.2 (or even provide the full Bayesian view).


3) Why does it make sense to use $b(t,v)$ as conditioning at intermediate time points, given the fact that $r(s,x)=∇ \log h(s,x) = ∇ \log \int p(T, y|t,x) q(v|y) \nu(dy)$.   Further, why is this meaningful when $v \in \mathbb{R}^{d^\prime}$ with $d^\prime < d$?

**Other Strengths And Weaknesses:**

See other comments

**Questions For Authors:**

4) Is the pCN used for experiments with the guided proposal?  I think this is difficult for the uninitiated reader to follow, particularly when it comes to Lines 347-355 R comparing the

5) If MCMC vs. no-MCMC is a distinction between the guided proposal and neural guided proposal, then this should be more clearly emphasized to distinguish the presented method and demonstrate its efficacy over the guided proposal.

**Relation To Broader Scientific Literature:**

The paper builds on the guided proposals of Schauer et. al 2017, Mider et. al 2021 using neural network control drifts.    The latter idea has been an active area of recent research in diffusion model finetuning and diffusion bridge literature, along with transition path sampling applications in computational chemistry.   However, these works often use the base drift of the original problem without including additional drift terms from the auxiliary, tractable bridge process.

Hence, I expect the paper to be interesting to the ICML community.

**Theoretical Claims:**

I would like to see more explicit reasoning to justify Eq. 5 and 10-12, just to give the reader more intuition for (i) the evolution of the auxiliary h-function and the (ii) appearance of particular terms in Eq 10 (particularly the Hessian).

I have confirmed the correctness of of the proposed method.

---

> ### Author Rebuttal · Authors · 2025-03-31
>
> **Claims and Evidence**
>
> The guided bridge relies on training the neural network. Once trained, independent samples are obtained. The quality of the guided proposal depends much on the nonlinearity of the diffusion and number of pCN-steps required in the MCMC-algorithm. This is case dependent. For the performance evaluation, please refer to the table in our reply to Reviewer 8sR1.
>
> **Methods and Evaluation Criteria**
> Inclusion of the guiding term from the guided proposal  is necessary to ensure the guided neural process will be a bridge. Without it, the neural net has to learn the unbounded term which behaves near the endpoint like $(v-x)/(T-t)$ (in case of a full observation and a uniformly elliptic diffusion).
>
> **Theoretical Claims**
> 1. A proof that $E_t$ in Eq. (5) is a local martingale is given in Palmowski & Rolski: it follows from partial integration and involves Dynkin's martingale. We believe this to be out of scope for this paper as it heavily depends on stochastic calculus.
> 2. For Eq. (10): the ``weight'' is obtained by integrating $$\frac{({\mathcal{A}}-\tilde{\mathcal{A}})\tilde{h}}{\tilde{h}}=\sum_i(b_i-\tilde{b}_{i})\frac{\partial_i\tilde{h}}{\tilde{h}}+\frac{1}{2}\sum_{i,j} (a_{ij}-\tilde{a}_{ij})\frac{\partial^2_{ij}\tilde{h}}{\tilde{h}}$$ over $[0, t]$.
> 3. Eq. (11) and (12) are taken from Section 2 of Mider et al.
>
> Due to the space limitation, we can't include more details about 2 and 3 in this reply, but we are happy to proceed them in the conversation.
>
> **Experimental Designas or Analyses**
>
> * "Cell dynamics": Yes, this is true. Empirical results show multimodal marginal distributions emerge after $t=2.0s$ (Fig. 6), with neural-guided bridges evolving primarily unconditionally beforehand. Peak splitting coincides with intensified drift forcing toward $v$, matching guided proposal dynamics. Neither approach fully replicates the three unconditional sampling modes, though all three become distinct during $t=2.0–3.0s$ under weakening constraints – despite potential intermediate convergence failures toward $v$.
>
> **Supplementary Material**
>
> We acknowledge the importance of additional experimental results, and will prioritize space allocation for key figures and tables while maintaining methodological rigor during the revision of the manuscript.
>
> **Essential References Not Discussed**
>
> We agree that Proposition is known in the literature. However, we did not find another reference where this is spelled-out for our purposes in detail. We simply want to have one clean statement that explains the change-of-measure to $\mathbb{P}^\star$ for this particular $h$-function, in the setting of SDEs.
>
> Thanks for pointing out the ``non-essential recent work'':
> 1. Denker et al.'s diffusion framework (Proposition 2.2) shares our Proposition 3.2's $h$-function concept (termed "generalized $h$-transform"), though methodologically distinct through denoising score matching. We will add reference to contextualize prior appearances of this construct.
> 2. While our focus remains on KL divergence's mode-seeking properties, we acknowledge log-variance divergence's potential benefits for mode collapse mitigation. This comparison will be expanded in future research.
> 3. The adjoint matching method shows promising scalability for high-dimensional applications (images/point clouds). We plan to investigate adapting our objective function into this framework.
>
> **Other comments or suggestions**
> 1. (a) We agree that it is useful to the reader to remind them of the definition of $b^\circ$ in Eq. (6). There is no $\tilde{b}^\circ$ in our paper.
>
>     (b) We agree with your suggestion and will add a remark at the end of Section 3.2 to explain this.
>
>     Indeed, if $\sigma$ is invertible, then for
>     $\tilde{X}_t = \sigma W_t$ transition densities $\tilde{p}$ are Gaussian and
>     $$ \nabla_x \log \tilde{p}(t,x; T,v) = (\sigma\sigma^T)^{-1} \frac{v-x}{T-t}.$$ Therefore, the bridge has drift $(v-x)/(T-t)$. We added this to the text.
> 2. We will clarify this in the revision of the manuscript. The following hierarchical model is meant: $$ v\mid y\sim q(v\mid y), y\sim p(T,y \mid 0,x_0).$$ Here, $y$ is the parameter that gets the prior $p( T,y \mid 0,x_0)$ assigned; $v$ is the observation. In this model, the likelihood can be written as $L(y; v)=q(v\mid y)$.
> 3. We are unsure what is meant by the remark on $b(t,v)$. Regarding the reason it is useful for $d'<d$: we may only observe certain components of the diffusion, for example, if we have a 2 dimensional example and observe $v \sim N(x_1, \sigma^2)$, then the dimension of the observation is lower than that of the diffusion. The Fitzhugh-Nagumo example is an illustration of this.
> 4. Yes, we will stress this in the revision and clarify the explanation given in lines 347-355. Indeed, for the guided proposal we always use MCMC with pCN-steps, contrary to the other methods.
> 5. We think we have addressed this comment in the previous item.

---

### Official Review · Reviewer_JEjz · 2025-03-11

**Overall Recommendation:** 3

**Summary:**

This paper introduces Neural Guided Diffusion Bridges, which added variational inference to Guided Proposal.
Neural-guided Diffusion Bridges show bridge sampling without MCMC. They perform competitively in many experiments. They can handle rare events, which is very hard for other methods.
Though such good performances, it has some limitations; it could become mode-seeking in multimodality, it occasionally samples from only a single mode in FitzHugh-Nagumo model's rare event, etc.


## update after rebuttal

The rebuttal clarified my concerns, and I continue to see this paper as leaning towards acceptance.

**Claims And Evidence:**

The claims are explicit using mathematical proof. Also, the claims are supported by appropriate experiments.

**Essential References Not Discussed:**

No

**Experimental Designs Or Analyses:**

In lines 402-405, the authors say, "The neural guided bridge samples paths from only one of the two modes, though the sampled paths appear very similar to the actual conditioned paths. "
Questions
1. Could we choose which modes to sample among the two modes?
2. Is there any reason for that result? I want to know the authors' thoughts

**Methods And Evaluation Criteria:**

The methods are explicit. In addition, the authors use various examples to demonstrate their methods' competitive performances in the experiments part.

**Other Comments Or Suggestions:**

No

**Other Strengths And Weaknesses:**

**Strengths**

This paper shows great math skills, resulting in neat development for the main idea.

**Weaknesses**

It would be better to show a realistic example, such as generated outputs(like images, video, and text) and metrics related to those outputs.

**Questions For Authors:**

Similar to **Experimental Designs or Analyses**,  is there any proof or reasoning for the experiments' result?
I see only results and explanations, except for proof or detailed reasoning about the experiments' result provided in the paper.
It would be better if you could give proof or reasoning about the experiments' result.

**Relation To Broader Scientific Literature:**

The contributions could be utilized for generative modeling, especially sampling.

Although sampling is very important for the generative models, it needs a hard cost for accurate sampling. If the contributions would be helpful to overcome those limitations,

**Theoretical Claims:**

I have one question for lines 271~272. How could the authors use 'local minimizer' and 'lower bound' at the same time? So, what I want to know is that, is there any condition, we could get a lower bound by choosing a local minimizer?

---

> ### Author Rebuttal · Authors · 2025-03-31
>
> **Theoretical Claims**
>
> The formulation was imprecise and we propose to reformulate it as follows:
> > If $\theta_{\rm opt}$ is a local minimizer of $L$ and $L(\theta_{\rm opt})=-\log \frac{\tilde{h}(0,x_0)}{h(0,x_0)}$, then $\theta_{\mathrm{opt}}$ is a global minimizer. This implies from which we obtain $D_{KL}(\mathbb{P}^{\bullet}_{\theta_{\mathrm{opt}} | \mathbb{P}^{\star})=0$, from which we obtain $\mathbb{P}^{\bullet}_{\theta_{\mathrm{opt}}} = \mathbb{P}^{\star}$.
>
> **Experimental designs or analyses**
>
> We cannot explicitly choose the target mode to sample from. Theoretically, the variational class $\{\mathbb{P}^{\bullet}_{\theta};\theta\in\Theta\}$ is large enough to contain the target $\mathbb{P}^{\star}$, and the global optimal $\mathbb{P}^{\bullet}_{\theta_{\mathrm{opt}}} = \mathbb{P}^{\star}$ can be guaranteed.
>
> However practically, the inherent non-convexity of the objective function induced by $G(s, x)$ predisposes $\vartheta_{\theta}$ to converge to local minima, as empirically evidenced by sample concentration within individual modes. However, directing the optimization trajectory within this complex landscape remains nontrivial, and we currently do not have a reliable method steer the process towards preferred local minima.
>
> **Questions for Authors**
>
> We don't fully understand what you mean by "It would be better if you could give proof or reasoning about the experiments' result.". We kindly ask you to clarify so we can answer your question accurately.

---

> > ### Comment · Reviewer_JEjz · 2025-04-02
> >
> > Reply for **Questions for Authors**:
> >
> > In "Experimental Designs or Analyses", I wrote "the neural guided bridge samples paths from only one of the two modes...".
> > I would like to understand why these two modes appear.
> >
> > Note for **Weaknesses**:
> >
> > The point I raised in the weaknesses section is just a suggestion.
> > While providing additional realistic examples could be helpful, I understand that it may not be easy within the rebuttal period.
> > I believe that using tables or graphs, as you’ve already done, should be sufficient.

---

> > > ### Author Response · Authors · 2025-04-04
> > >
> > > Thank you for your clarification.
> > >
> > > Consider the deterministic part of **unconditioned** FHN model:
> > >
> > > $\frac{dX_{t,1}}{dt} = F_1(X_{t})=\frac{1}{\chi}(X_{t,1} - X_{t,2} + s - X^{3}_{t,1})$
> > >
> > > $\frac{dX_{t,2}}{dt} = F_2(X_{t})=\gamma X_{t,1} + X_{t,2} + \alpha.$
> > >
> > > We now look into their fixed points by inspecting the conditions for $\frac{dX_{t,1}}{dt} = \frac{dX_{t,2}}{dt} = 0$, leading to the equations:
> > >
> > > $X_1 - X_2 + s - X_1^3 = 0,$
> > > $\gamma X_1 + X_2 + \alpha = 0$
> > >
> > > Substituting $X_2$ obtains:
> > >
> > > $X_1^3 - (1 + \gamma)X_1 - (s + \alpha) = 0,$
> > >
> > > which is a cubic equation, the root that solves this equation gives fixed points. The discriminant is $\Delta = 4(1+\gamma)^3 - 27(s+\alpha)^2$. Under the setting we considered in the paper, $[\chi, s, \gamma, \alpha, \sigma] = [0.1, 0, 1.5, 0.8, 0.3]$, $\Delta > 0$, suggesting there are three real roots. Explicitly, these three roots are $X_1=\\{1.72, -1.39, -0.34\\}$, and accordingly $X_2 =\\{-3.38, 1.28, -0.30\\}$, leading to three distinct fixed points. The stability of these points can be verified by inspecting the evaluations of the Jacobian $J$ of $[F_1, F_2]^T$ at $(X_1, X_2)$, which is defined by:
> > >
> > > $J = \begin{bmatrix} \frac{\partial F_1}{\partial X_{t,1}} & \frac{\partial F_1}{\partial X_{t,2}} \\\ \frac{\partial F_2}{\partial X_{t,1}} & \frac{\partial F_2}{\partial X_{t,2}} \end{bmatrix} = \begin{bmatrix} \frac{1-3X^2_{t,1}}{\chi} & -\frac{1}{\chi} \\\ \gamma & 1 \end{bmatrix},$
> > >
> > > and stability requires $\rm{Tr}(J)<0$ and $\rm{Det}(J)>0$, which turns out that all these three points are **unstable**.
> > >
> > > However, when conditioning on observations, the drift shall change. Assuming this change is known, we can apply the previous analysis to identify the fixed points and assess their stability. If some fixed points are stable, the diffusion term—though previously neglected—introduces noise around these points and enables transitions between their basins of attraction. The number of fixed points reflects the number of modes. In practice, however, the drift induced by conditioning is not available in closed form. As a result, we can only state that, under the rare event considered, the conditioned process appears to have exactly two stable fixed points. An alternative approach is to numerically approximate the Kolmogorov forward or backward equations to study the transition densities, as these densities are also not available in closed form either.
> > >
> > > We hope this could address your question.

---

### Official Review · Reviewer_8sR1 · 2025-03-18

**Overall Recommendation:** 3

**Summary:**

The paper presents a novel method for simulating conditioned diffusion processes, called diffusion bridges. This approach trains a neural network to approximate the bridge dynamics, providing a more robust and efficient alternative to traditional methods like MCMC or reverse-process modeling, particularly for rare events and multimodal distributions. By learning a flexible variational approximation of the diffusion bridge path measure, partly defined by a neural network, the method enables efficient independent sampling similar to simulating the unconditioned process. The paper validates this "neural-guided diffusion bridge" through various numerical experiments, comparing its performance against existing techniques in challenging scenarios.

**Claims And Evidence:**

yes, the claims made in the submission are supported by clear and convincing evidence.

**Essential References Not Discussed:**

All relevant references are discussed.

**Experimental Designs Or Analyses:**

1. The paper emphasizes that the proposed method learns directly from conditional samples, offering improved training efficiency compared to score-learning methods that rely on unconditional samples. Could you provide a more quantitative comparison of the training times and computational resources needed for your method versus the score-matching approach from (Heng et al., 2022) and the adjoint method from (Baker et al., 2024a) in one or more of the experimental settings?

2. The paper notes that the canonical score-matching loss involves inverting $\sigma\sigma^T$, which can be challenging for high-dimensional and hypo-elliptic diffusions. Could you elaborate on how your method avoids these particular challenges and how it may offer advantages in such scenarios?

**Methods And Evaluation Criteria:**

1. Could you provide more details on the specific neural network architecture used to parameterize the drift correction term v(t,x) in the different experiments? For example, what was the rationale behind selecting the number of hidden layers, the dimensionality, and the activation functions?

2. For the "Stochastic landmark matching" task, since the true bridge is intractable, it’s challenging to make a quantitative comparison. Are there alternative metrics or qualitative analyses, beyond visual inspection, that could help assess the performance of the proposed method relative to the guided proposal in high-dimensional settings?

**Other Comments Or Suggestions:**

The paper is well-written and easy to follow.

**Other Strengths And Weaknesses:**

Strengths:
Demonstrates robustness across different diffusion specifications and conditioning scenarios.

Effective in handling rare events and multimodal distributions.

Enables efficient independent sampling after training, with a cost similar to the unconditioned forward process.

Scales well to relatively high-dimensional problems, outperforming MCMC-based methods.

Weakness:
Although the neural-guided bridge and the guided proposal demonstrated comparable performance, both methods could only capture part of the modes in a multimodal distribution resulting from specific initial conditions. The paper notes that, while the neural bridge might not recover all modes, it offers faster sampling as a trade-off. It also suggests that additional MCMC updates could be applied to the trained neural bridges for better-quality samples.
The "FitzHugh-Nagumo model" experiment (Section 5.3), which involved a rare event, also highlighted a limitation in capturing all modes. Although the reference-conditioned process was bimodal, the neural-guided bridge only sampled paths from one of the two modes.

**Questions For Authors:**

1. In the Brownian bridge and Ornstein-Uhlenbeck bridge experiments, the lower bound of the loss function L(θ) could be computed analytically and used as a benchmark. For more complex, non-linear examples where this is not feasible, how do you ensure that the trained neural network is performing optimally and not converging to a suboptimal solution?

2. In Section 3.3, it is mentioned mention the importance of "matching conditions" for the linear auxiliary process, particularly for hypo-elliptic diffusions, to ensure P*≪ P. Could you elaborate on these conditions and explain how they were satisfied in the hypo-elliptic FitzHugh-Nagumo model discussed in Section 5.3?

3. In the "Multi-modality" experiment (Section 5.2), both the neural guided bridge and the guided proposal were able to cover only part of the modes. Considering the mode-seeking nature of variational inference, did you explore any strategies, or could you suggest approaches, to better capture the full multimodality of the conditioned process without relying on multiple MCMC chains?

4. Could you discuss the potential advantages or disadvantages of jointly learning $\vartheta_\theta$ and the parameters of the auxiliary linear process (e.g., $B$, $\beta$, $\sigmã$) using variational inference, as proposed for future work? What challenges do you anticipate in implementing this joint learning approach?

5. Section 6 suggests extending the approach to conditioning on partial observations at multiple future times. Could you outline the key modifications needed in your current methodology to accommodate such conditioning scenarios?

**Relation To Broader Scientific Literature:**

The proposed method aims to improve the efficiency of guided proposal-based simulation by replacing computationally expensive MCMC/SMC steps with a learned neural network, enabling fast, independent sampling. It also offers an alternative to score-learning methods that learn directly from conditional information, potentially leading to better performance in challenging scenarios like rare events.

**Theoretical Claims:**

yes, all the theoretical claims made by the authors are correct.

---

> ### Author Rebuttal · Authors · 2025-03-31
>
> **Methods and Evaluation Criteria**
> 1. All $\vartheta_{\theta}$ implementations use fully-connected networks with $(1+d)$-dimensional inputs (time $t$ and state $x$). Time integration varies by dimensionality: direct concatenation for low-dimensional systems (Brownian/OU/cell/FHN) versus sinusoidal time embeddings with feature-wise modulation (Perez et al.) in high-dimensional landmark processes. Architectures employ tanh activations to ensure Lipschitz continuity (critical for Assumption 4.1 via gradient clipping). Layer counts/hidden dimensions were determined via loss-minimizing parameter sweeps.
> 2. To our knowledge, no established quantitative metrics exist for high-dimensional performance evaluation. Only visual assessment by brute force unconditional sampling (cell and FHN)—prohibitively inefficient for rare events, are available. In contrast, real-world applications (e.g., image/video) permit rigorous benchmarking through domain-specific metrics like FID/IS scores used in translation tasks.
>
> **Experimental designs or analyses**
> 1. We took two representative examples to compare the computational cost of the methods considered in more detail in the following table.
>
> ||OU||Cell|||
> |-|-|-|-|-|-|
> |Methods|#Params|Time|#Params|Time|Complexity|
> |Adjoint Bridge|21969|93.53s|22114|2465.06s|$\mathcal{O}(d^3)$|
> |Score Matching|26353|335.96s|26498|1496.87s| $\mathcal{O}(d^3)$|
> |Neural Bridge|1341|29.21s|3362|186.02s|$\mathcal{O}(d^2)$|
>
> 2. Our proposed loss eliminates matrix inversion of $a=\sigma\sigma^T$. Precomputable terms $L(t)$, $M(t)$, and $\mu(t)$ are solved once before training, while $\tilde{r}(t,x)$ (Eq. 11b) and $G(t,x)$ (Eq. 10b) are computed during integration - all without requiring $a^{-1}$. This contrasts with the canonical score matching loss (Heng et al.) which contains $\Sigma(t,x)=a(t,x)$. Such inversions prove numerically unstable for near-singular $a$, and critically fail in our FitzHugh-Nagumo example where $a$ is singular. Our method remains applicable where score matching becomes undefined. To show more insight why the (neural) guided bridge works for hypo-elliptic diffusions, consider the SDE $$dY_t = \{b(t, Y_t) + \sigma(Y_t)f(t, Y_t)\}dt + \sigma(Y_t)dW_t,$$ and denote the law of $Y$ to be $\mathbb{Q}$. $\mathbb{P}^{\star}$ is absolutely continuous with respect to $\mathbb{Q}$ if there exists a bounded solution $\eta$ to the equation $$b^\star(t,x)- b(t,x)-\sigma(x) f(t,x) = \sigma(x) \eta(t,x).$$ Recall that  $b^\star(t,x)=b(t,x) + \sigma(x) \sigma(x)^T \nabla_x \log h(t,x)$. Then the preceding display can be rewritten to $$\sigma(x) \left( \sigma(x)^T\nabla_x \log h(t,x) -f(t,x)\right) = \sigma(x) \eta(t,x).$$ Hence, one can easily see such an $\eta$ exists by the specific form of the additional drift. In this way, we circumvent inversion of $\sigma$. This also explains why the additional drift in the neural guided bridge contains the $\sigma$ premultiplication.
>
> **Questions for Authors**
> 1. We cannot ensure this, similarly to other applications of Variational Inference. But some heuristic methods can be helpful in our case. For example, we can initialize the neural bridge differently, and test whether it learns the same drift term.
> 2. This "matching condition" appeared first in Theorem 1 of Schauer et al. This paper deals with the fully observed, uniformly elliptic case. It says that $\tilde\sigma$ should be $\tilde{a}(T)=a(T,v)$, see also our discussion in Section 3.3 of our paper. The claim that all examples in Section 5 satisfy the matching conditions needs adjustment. It is true, except for the Fitzhugh-Nagumo (FHN) model. Thank you for drawing our attention to this: we adjusted the formulation.
> The matching condition specifically refers to Assumption 2.4 in Bierkens et al. It consists of verifying 4 inequalities. As the diffusivity is constant, the 4th of the these is trivially satisfied. The first and third assumptions can be verified similar to Example 3.2 in Bierkens et al. (with $\Delta(t)=(T-t)^{-1}$). The second assumption in there is on the difference $b(t,x) -\tilde{b}(t,x)$. Inspecting the proof, it suffices that the second assumption only needs to hold true for $t=T$ and those $x$ for which $Lx=v$. With our choice of $\tilde b(t, x)$, it reads $b(T,x) -\tilde{b}(T,x)=0$, therefore the second assumption also suffices.
> 3. Yes, please refer to our reply to Review ysG7's Question 5.
> 4. We now simply choose $B$ and $\beta$ in a rather simple way to ensure the neural bridge will end up in the right point.  We don't see any direct complication from parametrising these functions by a neural net, apart from computational resources. Presently, it is unclear if the additional training time is worth the effort.
> 5. Guided proposals in the case of multiple partial observations are discussed in Mider et al. We would start from this approach, and simply add a drift term on each of the segments between observations times, as we propose in the present paper for only one segment.

---

### Official Review · Reviewer_ysG7 · 2025-03-19

**Overall Recommendation:** 4

**Summary:**

This paper introduces a novel variational method for simulating conditioned diffusion processes (diffusion bridges) by proposing a more expressive guided proposal framework of Schauer et al. (2017) with a learnable drift correction term parameterized by a neural network. By leveraging variational inference, the proposed method overcomes key limitations in existing approaches: guided-proposal-based methods require a careful, non-trivial choice of an auxiliary process and rely on computationally intensive MCMC/SMC updates, while score-learning-based methods often struggle with inadequate exploration of rare event regions and the numerical challenges of inverting nearly singular diffusivity matrices.

The method is validated on a diverse set of problems. The experiments, which include comparisons with state-of-the-art guided proposals, score matching, and adjoint-process methods, demonstrate that the neural guided bridge is more flexible and adaptable.

**Claims And Evidence:**

- The paper claims that its method can generate independent conditioned samples at a cost comparable to simulating the unconditioned process—thus avoiding computationally intensive MCMC or SMC updates—by employing a variational inference approach; **however, it does not provide a direct comparison of the neural network training time complexity with that of MCMC/SMC or score optimization methods.**
- Extensive experiments in Section 5—including tests on the Brownian bridge, Ornstein-Uhlenbeck process, a cell diffusion model, the FitzHugh-Nagumo system, and stochastic landmark matching —offer both quantitative and qualitative evidence that the proposed method is competitively robust compared to guided proposals, score matching, and adjoint-process approaches.

**Essential References Not Discussed:**

N/A

**Experimental Designs Or Analyses:**

The experimental design is comprehensive and well-structured, as it evaluates the proposed method across a variety of diffusion bridge scenarios—from simple one-dimensional cases (Brownian and Ornstein-Uhlenbeck bridges) to more complex, nonlinear, and high-dimensional problems such as cell diffusion, the FitzHugh-Nagumo model, and stochastic landmark matching.

The authors complement qualitative visual comparisons with quantitative analyses (e.g., tracking training loss curves and comparing empirical distributions against known lower bounds in simpler cases), which provides a strong basis for assessing the method's performance relative to guided proposals, score matching, and adjoint-process approaches.

**Methods And Evaluation Criteria:**

While I appreciate the thorough evaluation of the proposed method, I encourage the authors to provide a direct comparison of the neural network training time complexity with that of MCMC/SMC or score optimization methods.

**Other Comments Or Suggestions:**

Minor
L166- fix the Typo

**Other Strengths And Weaknesses:**

The idea presented in the paper is novel and innovative and is particularly relevant as it addresses key challenges in existing methods. Given the growing interest in diffusion-based models across machine learning community, the proposed method has significant practical relevance for applications requiring efficient sampling of conditioned stochastic processes.

**Questions For Authors:**

1. How crucial is the assumption that the stochastic process  X has smooth transition densities for the validity of your theoretical results, and what would be the impact of relaxing this assumption to discrete densities?
2. Why is the non-noisy case emphasized in the paper? The authors primarily discuss scenarios in their experiments where observations are noise-free, but I would appreciate insights into how their approach performs when observations contain noise.
4. Section 3.4:- Why did the authors opt to add an additional learnable term to the drift rather than attempting to learn the entire drift function, and do they have empirical evidence to support the claim that the guided proposal’s sample paths significantly deviate from the true conditioned paths?
5. Is it possible to use drift correction in existing approaches such as score optimization based methods? If yes, can the authors discuss the challenges associated with using drift correction in existing approaches.
6. Given that mode collapse is a well-known challenge for such problems, did you observe any instances of mode collapse during your optimization? If so, could the authors elaborate on the nature of these occurrences and describe the specific strategies or modifications that they implemented to mitigate mode collapse?

**Relation To Broader Scientific Literature:**

The paper highlights several challenges with current methods for simulating diffusion bridges. In Guided-Proposal-Based Methods, the authors note that a careful, often complex, choice of an auxiliary process is required. Moreover, when these methods are combined with MCMC or SMC updates, the computational cost can become prohibitive, especially for strongly nonlinear or high-dimensional diffusions.

Whereas, Score-Learning-Based Methods, the other popular approach, rely on samples from the unconditioned process, which often do not adequately cover regions corresponding to rare events, leading to suboptimal performance. Additionally, the canonical loss function in these methods involves inverting the matrix $\sigma\sigma^\top$, a task that is particularly challenging for hypo-elliptic and high-dimensional diffusions where the matrix may be nearly singular, thus complicating stable and accurate optimization.

Furthermore, the literature distinguishes these problems from related areas such as the diffusion Schrödinger bridge and neural SDEs, where the focus is on connecting fixed marginal distributions or modeling entire data trajectories under stochastic dynamics.

**Theoretical Claims:**

The main contribution of the paper is empirical where many ideas are borrowed from (and referred to) Bierkens et al. (2020). I have skimmed through the derivations in the appendix and they look correct to me.

---

> ### Author Rebuttal · Authors · 2025-03-31
>
> **Claims and Evidence / Methods and Evaluation Criteria**
> * While neural bridges and MCMC-guided proposals differ methodologically—complicating direct cost comparisons—their forward simulation costs are comparable: for example, in the landmark process ($d=100$), the forward simulation time is 9.81ms (neural) vs 6.85ms (guided). Training complexities diverge as  $O(Nd^3)$ (score-matching/adjoint) vs  $O(N^2d^2)$ (neural), where $N$ is the number of time steps. Due to the space limitation, we can not include benchmark results here, but please refer to our reply to Reviewer 8sR1, and we also have added a more comprehensive summary into the manuscript per suggestion.
>
> **Other Comments or Suggestions**
>
> We have fixed the equation, thank you!
>
> **Questions for authors**
> 1. We are unsure what you mean by ``discrete densities''. Proposition 3.2 requires $\nabla_x \log h(s,x)$ to be well defined. The simplest way to ensure this  is to assume existence of smooth transition densities. Note however that we can write $h(t,x)= \int q(v\mid y) P(T, d y\mid t,x)$, where $P(T, d y\mid t,x)$ is the Markov kernel. This expression also makes sense if the kernel does not admit (smooth) densities with respect to some dominating measure. Informally, the case of conditioning on a state without noise corresponds to taking $q(v\mid y)$ like a Dirac mass in $v$ and then the above display should be interpreted as $ h(t,x) = p(T,v\mid t,x).$
> Clearly, existence of $\nabla_x \log h(s,x)$ requries that $h$ is strictly positive and that its gradient exists. Throughout the paper we have assumed that the distribution of $v$ conditional on $y$ is Gaussian. It is however possible to relax this assumption.
> 2. The non-noisy case is most challenging. Intuitively, the larger the noise, the less the process needs to be guided in a certain direction. So we tested the approaches on the more difficult case.
> We formulated Proposition 3.2 deliberately for the non-noisy case;  in case we condition on the full state without noise, we can simply take $h(t,x) = p(T,v\mid t,x)$. However, in case of a partial observation, such as with the FitzHugh-Nagumo model, the form of $h$ reads more pleasantly when noise is assumed. We refer to Section 1.3.2. of Bierkens et al. for the somewhat more involved formulas that appear when no noise is assumed. So the motivation is merely to present a ``clean'' statement. In the examples, we assume $q(v\mid y) = \psi(v; L y, \epsilon^2 I)$, where $\epsilon$ is very small. For example, in the FHN-example $L=[1, 0]$, which corresponds to only observing the first component. Taking $\epsilon$ nonzero makes guided proposals also numerically better behaved near the time of conditioning.
> 3. The drift of the bridge behaves in a very specific way near the point we condition on. This is most easily seen in the uniformly elliptic case and when we condition on the full state $v$ at time $T$. Essentially, the drift of the true bridge behaves for $t\approx T$ as $(v-x)/(T-t)$. Failure to replicate this behaviour breaks absolute continuity between true and proposal bridge laws. Our guided proposal network explicitly replicates this asymptotic behavior. The neural bridge's auxiliary drift (confined to $\sigma$'s range and bounded) enhances proposals on $[0, T-\eta)$ for small $\eta$, preserving absolute continuity when added. Empirical validation appears as Fig. 6 in Bierkens et al.
> 4. In general yes. The additional learned drift in our method is the difference between the true score $\nabla_x\log h(s, x)$ and the proposed score $\nabla_x\log \tilde{h}(s, x)$ (with $\sigma$ as a scaling). The objective function is based on a closed-form expression for the  likelihood ratio between the target measure $\mathbb{P}^{\star}$ and the proposal measure $\mathbb{P}^{\circ}$  (As Eq. (10) in the paper). The key challenges in applying our method are to select $\mathbb{P}^{\circ}$ that satisfies two criteria: the likelihood ratio between the target $\mathbb{P}^{\star}$ and $\mathbb{P}^{\circ}$ is tractable, and sampling paths under $\mathbb{P}^{\circ}$ can be done efficiently.
> 5. Yes, mode collapse occurs in both cell diffusion and FitzHugh-Nagumo (FHN) experiments, showing multimodal marginal distributions. This stems from the forward KL divergence's mode-seeking objective. We've also considered reverse KL alternatives, but it will introduce problematic stochastic integrals and unstable optimization. Additionally, we see three possible mode collapse explanations: (i) local optimum convergence; (ii) misspecified endpoint guidance from $\tilde X$; (iii) Oversimplified $\vartheta$. We conjecture (i) to be most likely. We implemented strategies:  (i) optimizer/hyperparameter sweeps (Adam/RMSprop/SGD); (ii) dropout regularization. Neither approach yielded significant improvements. We hypothesize that the presence of $G(t,x)$ in the objective function introduces significant non-convexity, rendering the optimization landscape inherently challenging.

---

### Decision · Program_Chairs · 2025-05-01

**Decision:**

Accept (poster)

**Comment:**

The reviewers agree that this is an interesting and solid paper addressing an important and topical problem. The consensus is that this would make a useful addition to ICML. The reviewers have made a number of suggestions to improve the clarity of the work, and the authors have also offered some changes/additional empirical results. Please go carefully through the discussion below and incorporate these changes before the camera-ready version is due.